# Miro clusters regulate ER-mitochondria contact sites and link cristae organization to the mitochondrial transport machinery

Souvik Modi [1,2,8]*, Guillermo López-Doménech [1,8]*, Elise F. Halff [1,7], Christian Covill-Cooke[1],
Davor Ivankovic [1], Daniela Melandri [1], I. Lorena Arancibia-Cárcamo [1], Jemima J. Burden [3],
Alan R. Lowe [4,5,6] & Josef T. Kittler [1]*

Mitochondrial Rho (Miro) GTPases localize to the outer mitochondrial membrane and are essential machinery for the regulated trafficking of mitochondria to defined subcellular locations. However, their sub-mitochondrial localization and relationship with other critical mitochondrial complexes remains poorly understood. Here, using super-resolution fluorescence microscopy, we report that Miro proteins form nanometer-sized clusters along the mitochondrial outer membrane in association with the Mitochondrial Contact Site and Cristae Organizing System (MICOS). Using knockout mouse embryonic fibroblasts we show that Miro1 and Miro2 are required for normal mitochondrial cristae architecture and Endoplasmic Reticulum-Mitochondria Contacts Sites (ERMCS). Further, we show that Miro couples MICOS to TRAK motor protein adaptors to ensure the concerted transport of the two mitochondrial membranes and the correct distribution of cristae on the mitochondrial membrane. The Miro nanoscale organization, association with MICOS complex and regulation of ERMCS reveal new levels of control of the Miro GTPases on mitochondrial functionality.

[1] Neuroscience, Physiology and Pharmacology, University College London, Gower Street, WC1E 6BT London, UK. [2] Department of Biological Sciences, Tata Institute of Fundamental Research, Homi Bhabha Road, Mumbai 400005 Maharashtra, India. [3] MRC Laboratory for Molecular Cell Biology, University College London, Gower Street, WC1E 6BT London, UK. [4] Structural & Molecular Biology, University College London, Gower Street, WC1E 6BT London, UK. [5] Department of Biological Sciences, Birkbeck College London, WC1E 7H London, UK. [6] London Centre for Nanotechnology, 17-19 Gordon Street, WC1H 0AH London, UK. [7] Present address: Department of Psychosis studies, Institute of Psychiatry, Psychology and Neuroscience, King's College London, 16 De Crespigny Park, SE5 8AB London, UK. [8] These authors contributed equally: Souvik Modi, Guillermo López-Doménech. *email: souvikmodi@gmail.com; g.lopez-domenech@ucl.ac.uk; j.kittler@ucl.ac.uk

Mitochondria generate ATP to drive key cellular functions, including ion pumping, intracellular trafficking and cellular signaling cascades[1,2]. The mitochondrial population are trafficked to where they are needed to meet local energy and $Ca^{2+}$ buffering demands[3]. Miro proteins form complexes with the TRAK adaptors and dynein/kinesin motors to regulate the microtubule-dependent transport of the mitochondria[4,5]. Recently, an actin-dependent transport of the mitochondria has also been linked to Miro regulation through the recruitment and stabilization of the mitochondrial myosin 19 (Myo19) to the outer mitochondrial membrane (OMM)[6,7]. In yeast, Miro exists as a single orthologue, Gem1, important for correct mitochondrial inheritance and cellular viability[8,9]. In mammals, there are two Miro family members, Miro1 and Miro2, that share ~60% sequence identity, comprising two GTPase domains flanking two EF-hand $Ca^{2+}$-binding domains and a C-terminal transmembrane domain that targets them to the OMM[10]. Although their role in mitochondrial transport is well established, far less is known about their interactions with other key protein complexes located at the OMM or the inner mitochondrial membrane (IMM).

The mitochondrial contact site and cristae organizing system (MICOS) located in the IMM is a large protein complex (often > 1 MDa) mainly formed by Mic60/Mitofilin, Mic19/CHCHD3, Mic10/MINOS1, and Mic25/CHCHD6[11–14]. Constituents of the MICOS complex are concentrated in discrete patches on the IMM decorating the cristae junctions[15]. The MICOS complex is crucial for maintaining cristae architecture as knockdown of MICOS components leads to mitochondria with altered cristae morphology resulting in compromised oxidative phosphorylation[16,17]. In addition, the MICOS complex acts as a bridge between the OMM and the IMM by forming a higher order complex with Sam50, known as the mitochondria intermembrane bridging complex (MIB)[18]. Although it is postulated that yeast Gem1, and *Drosophila* dMiro, could be associated with individual MICOS components[19,20], association of mammalian Miro proteins with intact MICOS complex and its functional role has not yet been characterized.

Mitochondria also engage in physical interaction with the endoplasmic reticulum (ER) through dedicated protein complexes at contact sites, known as ERMES (ER–Mitochondria Encounter Structures) in yeast or ER–Mitochondria contact sites (ERMCS) in mammals[21]. Yeast Gem1 and *Drosophila* dMiro have been identified as integral parts of the ERMES and ERMCS complexes, respectively[8,22]. Interaction mapping in yeast established that ERMES components and MICOS complex genes shared a strong genetic interaction between them and also identified similar interactions with *gem1*[19]. However, the relationship between Miro proteins and the MICOS and ERMCS complexes in mammalian cells remains largely unexplored.

By combining biochemical, super-resolution, and electron microscopy techniques, we address here the roles of mammalian Miro proteins in the regulation of mitochondrial cristae architecture and ERMCS in mammalian cells. Genetic ablation of both Miro proteins in mouse embryonic fibroblasts (MEF) results in reduced ERMCS and in the disruption of mitochondrial cristae organization. Using Structured Illumination Microscopy (SIM) and dual-color direct Stochastic Optical Reconstruction Microscopy (dSTORM) imaging of the mitochondria, we show the submitochondrial organization of Miro proteins. Miro1 and Miro2 form discrete clusters on the mitochondrial membrane, the distribution of which closely correlate with MICOS components. Biochemically, we show that Miro1 and Miro2 interact with Sam50 and MICOS. Furthermore, we also show that Miro proteins link MIB/MICOS complexes, spanning the inner and OMMs, to the motor adaptor proteins TRAK1 and TRAK2. Our data establish a role for Miro proteins in connecting the mitochondrial transport machinery with the MICOS complexes to ensure the coordinated transport of both mitochondrial membranes and the homogeneous distribution of cristae and cristae junctions inside the mitochondria. Miro proteins, thus, guarantee the appropriate supply of the mitochondrial compartments responsible for energy production to the regions in the cell where mitochondria are delivered.

## Results

**Loss of Miro1 and Miro2 alters mitochondrial ultrastructure.** We recently showed that loss of both Miro proteins in MEFs leads to altered mitochondrial distribution and morphology[6]. To further explore the effects of Miro loss on mitochondrial structure, we performed Structured Illumination Microscopy (SIM), a super-resolution technique that provides an optical resolution almost twice the diffraction limit[23]. Using SIM, wild-type (WT) cells expressing a mitochondrial matrix-targeted GFP (mtRo^GFP) showed a predominance of thin and long mitochondria, with individual mitochondria having continuous GFP staining with occasional cells presenting short and round mitochondria (Fig. 1a, b). In contrast, ~70% of Miro1/2 double-knockout (DKO) cells presented a discontinuous and often hollow matrix-targeted GFP signal (Fig. 1a, b), which correlated with a predominance of shorter mitochondria with enlarged and more rounded mitochondrial segments. The discontinuous mitochondrial matrix suggests a role for Miro proteins in maintaining the architecture of the IMM. Importantly, re-expression of either ^mycMiro1 or ^mycMiro2 in DKO cells rescued mitochondrial matrix continuity (Fig. 1b; Supplementary Fig. 1).

To further explore the effects of depleting Miro proteins on mitochondrial structure and morphology, we carried out ultrastructural analysis of MEF cells using transmission electron microscopy (TEM). In WT MEFs, the majority of mitochondria showed normal cristae structure and a homogeneous cristae distribution throughout the mitochondrial segments (Fig. 1c–e). In contrast, the majority of DKO cells presented an altered cristae architecture with frequent vesiculated mitochondrial matrix (Fig. 1c, d). Importantly, DKO cells showed a nonuniform arrangement of the mitochondrial cristae, with some mitochondrial regions having the normal density of cristae alternated with regions that appeared enlarged and devoid of cristae (Fig. 1e). Together, these data indicate that Miro proteins play a role in regulating mitochondrial morphology and in the maintenance of the mitochondrial cristae architecture.

To test whether these changes in the mitochondrial structure could be a consequence of reduced levels of protein components known to regulate cristae structure, we carried out western blot analysis of lysates from three different WT and DKO MEF lines. We did not observe any significant change in the levels of several MICOS components tested (Mic19/CHCHD3, Mic60/Mitofilin, and Mic27/ApooL). The OMM protein Sam50, known to closely associate to the MICOS complex to form the MIB complex and bridge the IMM and the OMM, also did not show any significant change in DKO cells. In addition, we observed no changes in the levels of proteins related to Miro transport function (TRAK1 or TRAK2) or mitochondrial components, like ATP5α (CVα) or Tom20 (Fig. 1f). In contrast, we report a striking fourfold increase in the levels of Inositol 1,4,5-trisphosphate (IP3) receptor (IP3R), a known regulator of the ER and mitochondria contact sites that regulates $Ca^{2+}$ communication between the two organelles (Fig. 1f). Other components of ERMCS that act in coordination with IP3R, like GRP75 and VDAC1, were also found to be moderately upregulated, although not to a statistically significant level (Fig. 1f).

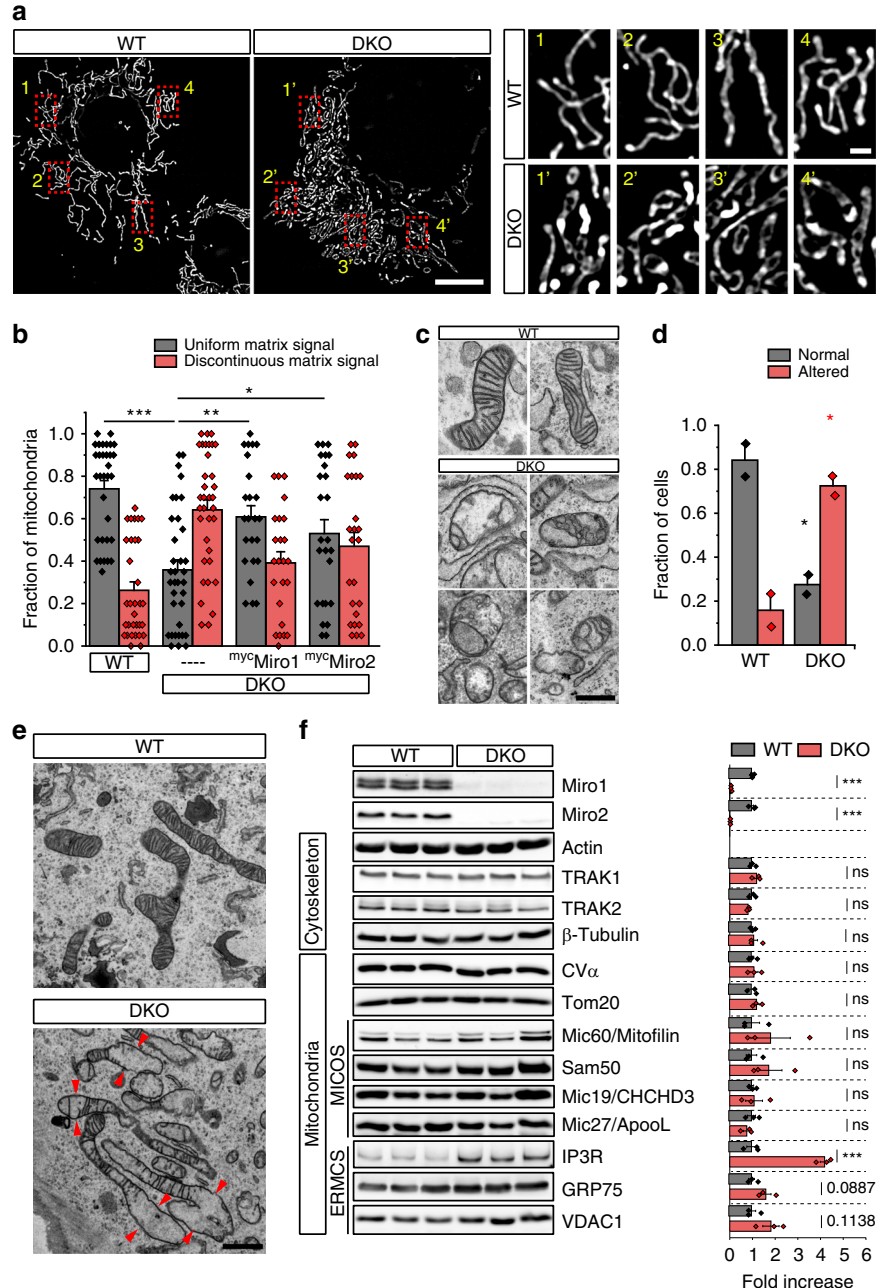

**Fig. 1** Loss of Miro is associated with altered cristae morphology. **a** Imaging of the mitochondrial matrix with mtRo$^{GFP}$. WT and Miro DKO MEF cells were transfected with the mitochondrial matrix-targeted mtRo$^{GFP}$ and imaged using structured illumination microscopy (SIM) (scale bar: 10 μm; insets: 1 μm). **b** Quantification of the images shown in (**a**) by scoring abnormalities in matrix continuity revealed by GFP in WT, DKO, and DKO cells re-expressing Miro proteins ($n$ = cells; in which 32 WT, 35 DKO, 24 DKO re-expressing Miro1 and 25 DKO re-expressing Miro2 cells were assessed, obtained from three independent preparations; One-way ANOVA, Bonferroni post hoc). **c** TEM images of mitochondrial cristae morphologies observed in WT and Miro DKO MEFs (scale bar: 500 nm). **d** Quantification of TEM images after classification of the cells as having normal cristae morphology or an altered cristae morphology ($n$ = experiments; in which 54 WT and 51 Miro DKO cells were analyzed from two independent sample preparations; Student's $t$ test with Welch's correction). **e** Representative EM images of the mitochondria from WT and DKO cells showing the homogeneity of cristae in WT cells and the appearance of spaces and enlargement of mitochondrial units in regions without cristae in DKO cells (scale bar: 1 μm). **f** Western blot analysis and quantification of three different cell lines independently generated for each genotype ($n$ = independently generated cell lines; three for WT and three for DKO; Student's $t$ test) to analyze cellular levels of proteins related to the cytoskeleton, MICOS complex, and ERMCS. Error bars represent ± SEM. Significance: *$p < 0.05$; **$p < 0.01$; ***$p < 0.001$

**Loss of Miro1 and Miro2 alters ER/mitochondria communication.** IP3Rs located at the ER are one of the main Ca$^{2+}$-release channels[24] and upon activation by IP3 can transfer Ca$^{2+}$ to the mitochondria through a IP3R–VDAC complex[25]. In mammalian cells, IP3R forms a complex with GRP75 and VDAC to maintain ERMCS[26]. At steady state, IP3R levels are tightly regulated, and alteration of IP3R expression has been implicated with changes in ER morphology and ER Ca$^{2+}$ release[27,28]. In addition, recent studies have demonstrated the presence of Gem1/dMiro at ER–mitochondria contact sites in yeast and *Drosophila*,

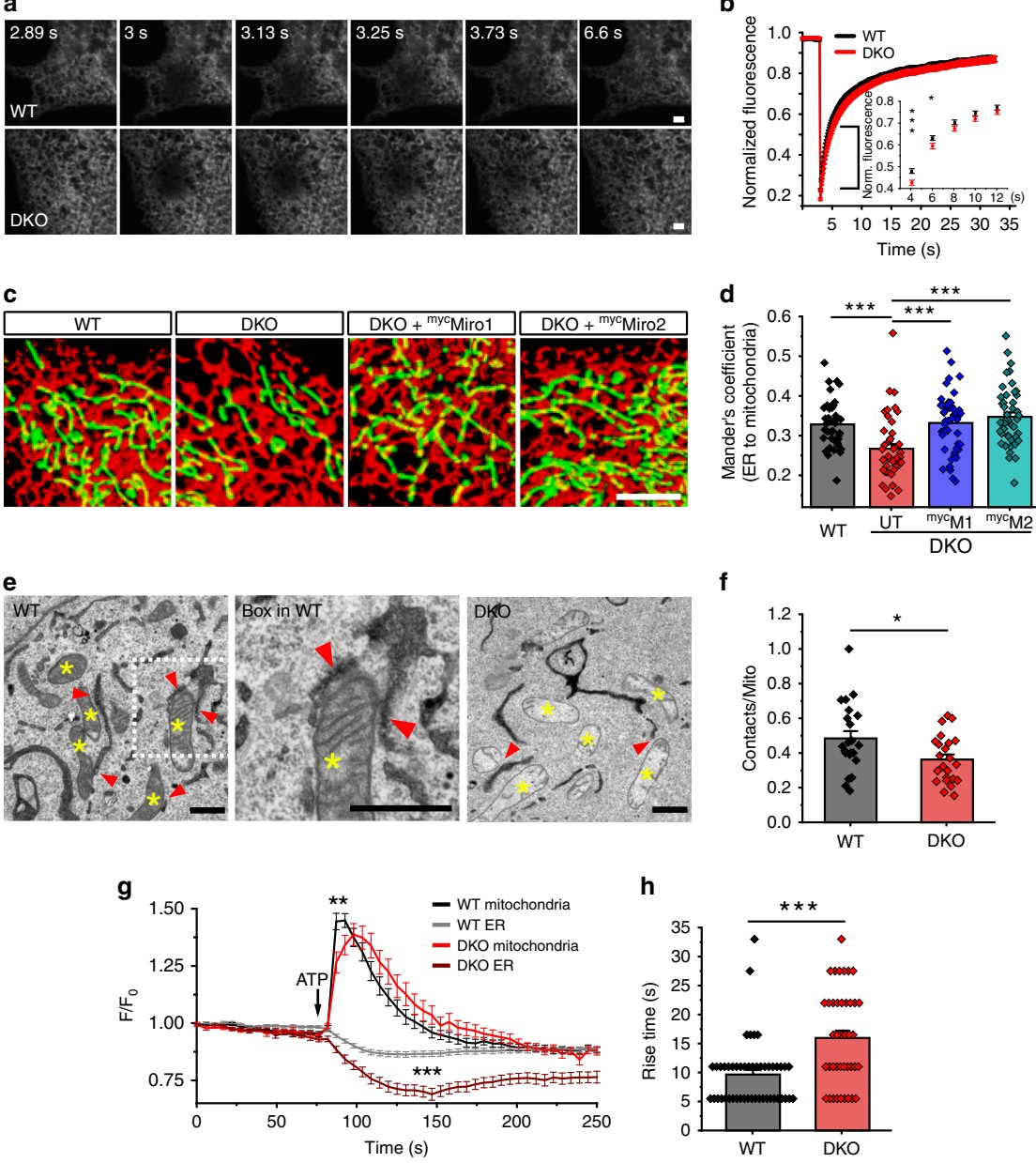

**Fig. 2** Miro control ER/mitochondria communication by regulating the number of ERMCS. **a** FRAP analysis of ER dynamics measured in MEF cells transfected with $^{DsRed}ER$ (scale bar: 2 μm). **b** Quantification of images shown in (**a**). Inset: expanded region of first 12 s of the recovery after bleaching (fraction of recovery at 4 s: WT ~0.47 ± 0.01, DKO ~0.42 ± 0.01, $p = 0.0010$; and after 6 s: WT ~0.63 ± 0.01, DKO ~0.59 ± 0.01, $p = 0.0311$; Student's $t$ test with Welch's correction; $n$ = cells; in which 37 WT and 38 DKO cells were analyzed from three independent experiments). **c** Representative images of MEF cells expressing $^{GFP}Su9$ and $^{DsRed}ER$ (scale bar: 10 μm). **d** Quantification of Mander's coefficient between the ER and mitochondria from (**c**) ($n$ = cells; in which 40 WT, 45 DKO, 47 DKO re-expressing Miro1, and 46 re-expressing Miro2 cells were used from three independent experiments; one-way ANOVA, Bonferroni post hoc). **e** Electron micrographs of the mitochondria and ER in WT and DKO cells expressing an HRP construct fused to an ER retention signal (KDEL). Yellow asterisks depict the mitochondria; red arrows point to the ER/mitochondria close contacts (< 35 nm). A magnified view from the WT image depicts close contacts between the ER and mitochondria (scale bar: 200 nm). **f** Quantification of ERMCS in TEM images ($n$ = cell; in which 22 WT and 23 DKO cells were used, from two independent sample preparations; Student's $t$ test with Welch's correction). **g** Agonist induced $Ca^{2+}$ release from the ER and subsequent mitochondrial $Ca^{2+}$ uptake. Arrow indicates addition of agonist ATP ($n$ = 49 WT and 41 DKO cells for the mitochondria and $n$ = 47 WT and 39 DKO cells for ER were analyzed from five independent experiments; Student's $t$ test with Welch's correction). **h** Rise time (calculated from baseline to maximum amplitude after addition of ATP) in WT and DKO cells ($n$ = cells; in which 49 WT and 41 DKO cells from five independent experiments were analyzed; Mann–Whitney's $U$ test). Error bars represent ± SEM. Significance: *$p < 0.05$; **$p < 0.01$; and ***$p < 0.001$

respectively[8,22,29]. To study whether Miro accomplishes a role in regulating the connectivity of the ER membranes, we performed a fluorescent recovery after photobleaching (FRAP) assay in our MEF cell lines. FRAP analysis of ER-luminal-targeted DsRed ($^{DsRed}ER$) showed no significant difference in fluorescence recovery between WT and DKO cells (Fig. 2a, b; mobile fraction $WT_M$ ~87 ± 7.5% against $DKO_M$ ~86 ± 8.2%, Mean ± SD; Student's $t$ test), with just a delay in the initial recovery time (Fig. 2a, b; $t_{1/2}$ recovery time: 1.80 s ± 1.56–2.4 s for WT and 2.40 s ± 1.80–3.46 s for DKO; median ± interquartile range (IQR), Mann–Whitney

$U$ test, $p = 0.019$). This indicates that organelle connectivity of the ER remains overall constant upon loss of Miro.

To more specifically test whether the increase in IP3R associates with changes in ERMCS in Miro DKO cells, we transfected WT and DKO cells with $^{GFP}$Su9 and $^{DsRed}$ER to label the mitochondria and ER, respectively, and carried out deconvolution confocal microscopy. DKO cells showed a significant decrease in the overlapping area between the ER and mitochondria (measured by Mander's coefficient) (Fig. 2c, d), which was specific to the loss of Miro as re-expression of either $^{myc}$Miro1 or $^{myc}$Miro2 in DKO cells rescued the amount of overlap between both compartments (Fig. 2c, d). This was further confirmed using TEM by transfecting an ER-targeted HRP construct (KDEL-HRP)[30] to enhance the contrast of ER structures and allow the identification and quantification of ERMCS (defined by proximity of the ER and mitochondria within 35 nm). DKO cells showed a decreased number of contacts between the two organelles (Fig. 2e, f), confirming that Miro proteins accomplish a role in regulating the ER and mitochondria association. In addition, we measured the mitochondrial Ca$^{2+}$ uptake upon ATP-induced Ca$^{2+}$ release from ER stores (Fig. 2g). WT cells required 9.65 s ($9.65 \pm 0.79$ s, mean ± SEM) to reach maximum amplitude (F/F$_{Min}$ = $1.44 \pm 0.03$-fold mean ± SEM), while DKO cell showed a significantly delayed uptake ($15.96 \pm 1.27$ s, mean ± SEM) and reduced amplitude (F/F$_{Min}$ = $1.26 \pm 0.04$, mean ± SEM) (Fig. 2g, h). This suggests that the ER/mitochondrial handling of Ca$^{2+}$ is severely affected as a consequence of a decrease in ERMCS in DKO cells. We also observed that upon treatment with ATP, there was a significantly larger loss of Ca$^{2+}$ from ER stores in DKO cells than in WT cells (Fig. 2g). This is probably due to increased level of IP3R receptors in DKO cells, as more IP3R may result in a significantly larger Ca$^{2+}$ release upon stimulation[28].

**Miro proteins associate with MICOS components.** Mitochondrial cristae are maintained by the interplay between the MICOS complex located at cristae junctions and Sam50 located at the OMM[31,32]. Alteration of the MICOS complex proteins has revealed their importance for the maintenance of mitochondrial cristae ultrastructure[32–34]. The disruption of cristae architecture in DKO MEFs (Fig. 1c, e) indicates a possible link between MICOS and Miro. Indeed, immunoprecipitation of $^{GFP}$Miro1 or $^{GFP}$Miro2 in HeLa cells revealed robust interactions with the core components of the MICOS complex Mic60/Mitofilin and Mic19/CHCHD3 and with the MIB complex-forming component Sam50, whereas an unrelated OMM protein, Tom20, did not co-immunoprecipitate with either Miro1 or Miro2 (Fig. 3a). Interestingly, we did not detect interaction between Miro and Mitofusins or Mtx1, suggesting that these interactions might be low-affinity, transient, or cell-type dependent (Fig. 3a). Control experiments with EGFP or the mitochondrially targeted $^{GFP}$Su9 did not co-immunoprecipitate any of the MICOS components tested (Fig. 3a; Supplementary Fig. 2) confirming the specificity of the interactions. Importantly, we confirmed endogenous association between Miro2 and Sam50 using specific antibodies against Miro2 in WT mouse brains, while as expected, anti-Miro2 antibodies did not co-immunoprecipitate Sam50 in brain lysates from Miro2 KO animals (Fig. 3b). Furthermore, Mic60/Mitofilin was also specifically co-immunoprecipitated with Miro2 in lysates from WT brains (Fig. 3b). We further confirmed the interaction of endogenous Miro2 with Sam50 and Mic60/Mitofilin in situ using a proximity ligation assay (PLA), which allows to test association of proteins that reside in close proximity (30–40 nm) within the same complex[35]. The presence of the antibody pair showed a fourfold enrichment of fluorescent signals compared

with the single antibodies confirming that native Miro2 and Sam50 as well as Miro2 and Mic19/CHCHD3 can be found associated (Fig. 3c–e). Thus, these experiments indicated that Miro proteins can be detected in a complex with key components of the MIB/MICOS complex.

**Miro localizes in discrete clusters on the mitochondrial surface.** Super-resolution imaging has shown that the MICOS complex can form an array that appears as "discontinuous rail-like" structures[15]. Since Miro proteins interact with several MICOS components and loss of Miro results in cristae deformation, we investigated the sub-mitochondrial organization of Miro proteins and its relationship with the MICOS complexes. Diffraction-limited confocal microscopy revealed that either GFP-tagged or myc-tagged Miro proteins expressed in HeLa cells localized to the mitochondrial network as expected (Fig. 4a). Using SIM imaging, we observed that expression of both $^{myc}$Miro1 and $^{myc}$Miro2 exhibited a discontinuous staining pattern on the mitochondrial membrane that was enriched in certain locations (Fig. 4a; Supplementary Fig. 3). Importantly, endogenous Miro2 in MEFs showed a similar discontinuous pattern, confirming that native Miro proteins localize in discrete domains along the mitochondrial membrane (Supplementary Fig. 4). To investigate this further, we took advantage of dSTORM[36,37] imaging (a super-resolution technique which provides almost sixfold higher resolution than SIM) and performed correlative SIM/dSTORM[38]. Using dSTORM, we observed individually resolved clusters in $^{myc}$Miro1 expressing HeLa cells along the mitochondrial membrane that were not resolved under SIM imaging (Supplementary Fig. 5). Similar nanoclusters were also observed upon dSTORM imaging of $^{GFP}$Miro1 or $^{GFP}$Miro2 in HeLa cells (Fig. 4b), while in contrast, mitochondrial matrix-targeted $^{GFP}$Su9 showed a more uniform distribution (Fig. 4b). In addition, we imaged HeLa cells expressing low to very high amounts of $^{GFP}$Miro2. Cluster analysis based on a pairwise correlation method[39] showed that Miro protein levels in the OMM do not play a significant role in this nanoscale organization (Supplementary Fig. 6A, B). Furthermore, we performed density-based spatial clustering of applications with noise (DBSCAN), which has been widely used to asses clustering of various membrane proteins[40]. The DBSCAN cluster map showed nanoscale domains formed by both $^{GFP}$Miro1 and $^{GFP}$Miro2 in HeLa cells (Fig. 4c). We next quantified the sizes of Miro clusters by two complementary methods. First, we analyzed the cluster sizes using Ripley's K-function[41] followed by quantification of cluster sizes post reconstruction of dSTORM images using the Feret's diameter (longest distance between any two points along the perimeter of each cluster). Ripley's K-function indicated that both $^{GFP}$Miro1 and $^{GFP}$Miro2 formed a cluster size around 100–150 nm (Fig. 4d). The distribution of diameters revealed clusters ranging from 50 to 250 nm (Fig. 4e). $^{GFP}$Miro1 clusters were found to have a median diameter ~100 nm, very similar to that of $^{GFP}$Miro2 clusters (Fig. 4e), both of which were much larger than the localization precision of the instrument (marked as a red bar in Fig. 4e). Finally, we also imaged $^{GFP}$Miro2 clusters in our MEF cells and in primary cultures of hippocampal neurons. We observed nanocluster-like organization of $^{GFP}$Miro2 in both MEFs and in primary hippocampal neurons similar to HeLa cells (Supplementary Fig. 7A, B), demonstrating that the nanoscale organization of Miro protein complexes appears to be conserved across cell types.

The cluster organization of both Miro1 and Miro2 prompted us to hypothesize that both Miro proteins could be interacting partners. By co-transfecting myc-tagged and GFP-tagged versions of Miro1 and Miro2 and performing GFP-trap

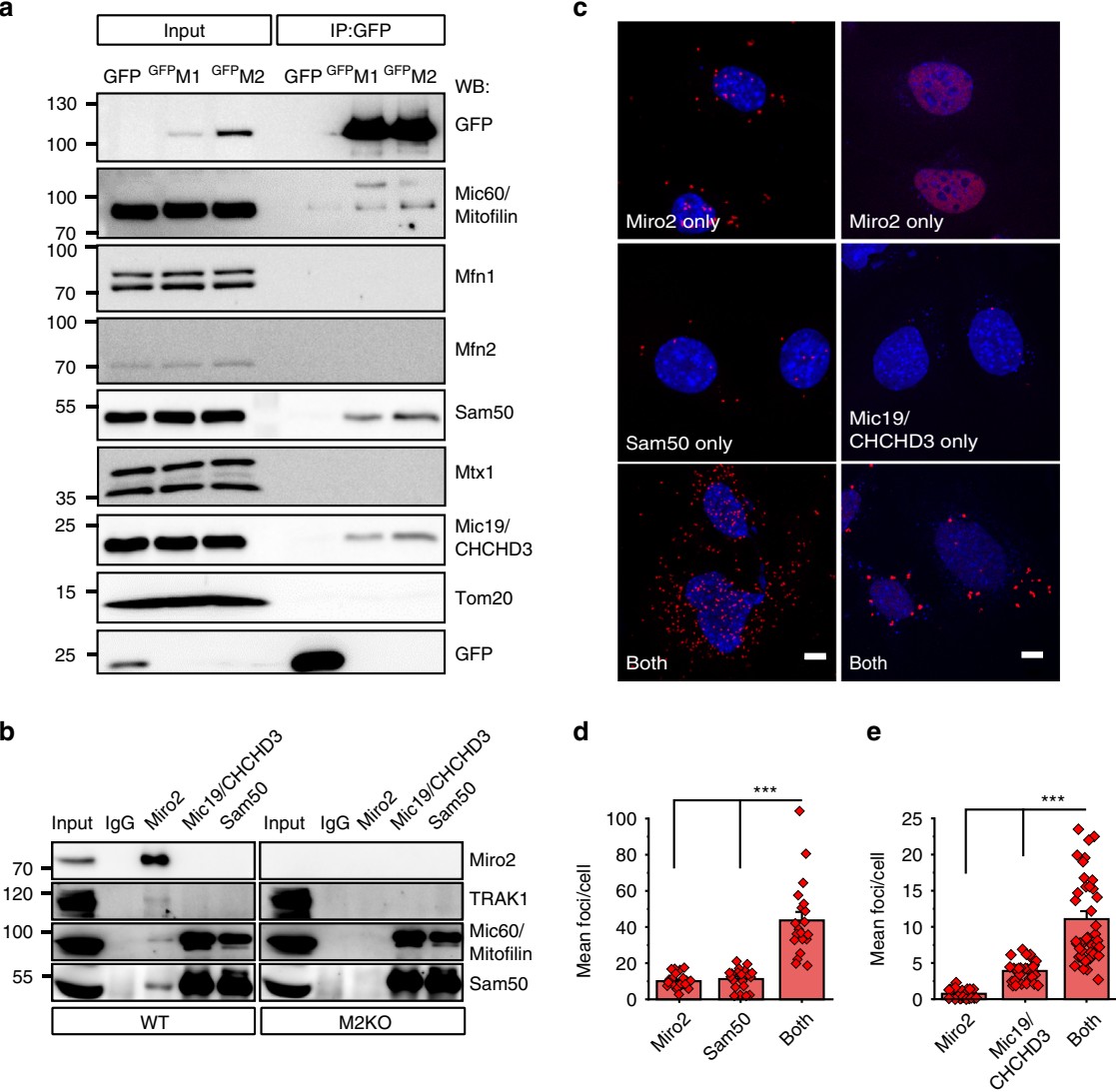

**Fig. 3** Miro proteins are a component of the MICOS complex. **a** Miro proteins interact with MICOS complex proteins. Western blot analysis of $^{GFP}$Miro protein complexes from HeLa cells were analyzed in SDS-PAGE for several MICOS components. **b** Co-immunoprecipitation of Miro2 with endogenous MICOS complex proteins. Immunoprecipitation was performed from WT and Miro2 KO brains, and western blot was performed with different antibodies against MICOS complex proteins. Approximately 2% of total cell lysate used for immunoprecipitation was loaded for inputs. **c** Proximity ligation assay (PLA) between Miro2 and OMM protein Sam50 (left panel) as well as PLA between Miro2 and the MICOS-specific protein Mic19/CHCHD3 (right panel). The nucleus is shown in blue (DAPI), and PLA assay products are shown in red (scale bar: 5 μm). **d**, **e** Quantification of the experiments shown in (**c**); (for (**d**) $n = 23$ images with only Miro2 antibody, 20 images for only Sam50 antibody, and 20 images with both antibodies and for (**e**) $n = 27$ images with only Miro2 antibody, 29 images for only Mic19/CHCHD3 antibody and 44 images with both antibodies, both from three independent experiments; One-way ANOVA, Bonferroni post hoc). Error bars represent ± SEM. Significance: ***$p < 0.001$

immunoprecipitations, we observed that both Miro1 and Miro2 can interact with themselves and with each other (Fig. 4f), suggesting that Miro1 and Miro2 multimers may form molecular platforms in the OMM upon which other mitochondrial molecular structures can be built.

**Miro nanodomains are associated with MICOS nanoclusters.** In conventional confocal imaging, Miro and MICOS components exhibit homogeneous staining appearing to co-localize with each other (Supplementary Fig. 8A, B). To explore whether MICOS clusters are closely associated with Miro nanodomains, we performed dual color dSTORM in HeLa cells transfected with $^{GFP}$Miro2 and immunostained with an antibody against Mic60/Mitofilin. Mic60/Mitofilin nanoclusters are regularly spaced and

sparser than those observed for Miro, however, both sets of clusters appeared similar in size (Fig. 5a). Mic60/Mitofilin and Miro2 clusters were often present in close proximity to one another with partial overlap between them. The extent to which the two proteins co-cluster was calculated using Van Steensel's cross-correlation function (CCF)[42] which showed a positive peak (Fig. 5d, e), indicating that both the clusters are positively correlated in the spatial domain. To test whether this correlation was specific, we transfected $^{GFP}$Miro2 and labeled with an anti-GFP antibody and a nanobody against GFP conjugated to Alexa 647. The correlation analysis and mean CCF values of both anti-GFP signals showed a similar pattern to that obtained for Mic60/Mitofilin indicating that the correlation between Miro2 and Mic60/Mitofilin is specific (Fig. 5b, d, e). Similar results were obtained when we calculated the nearest neighbor distance

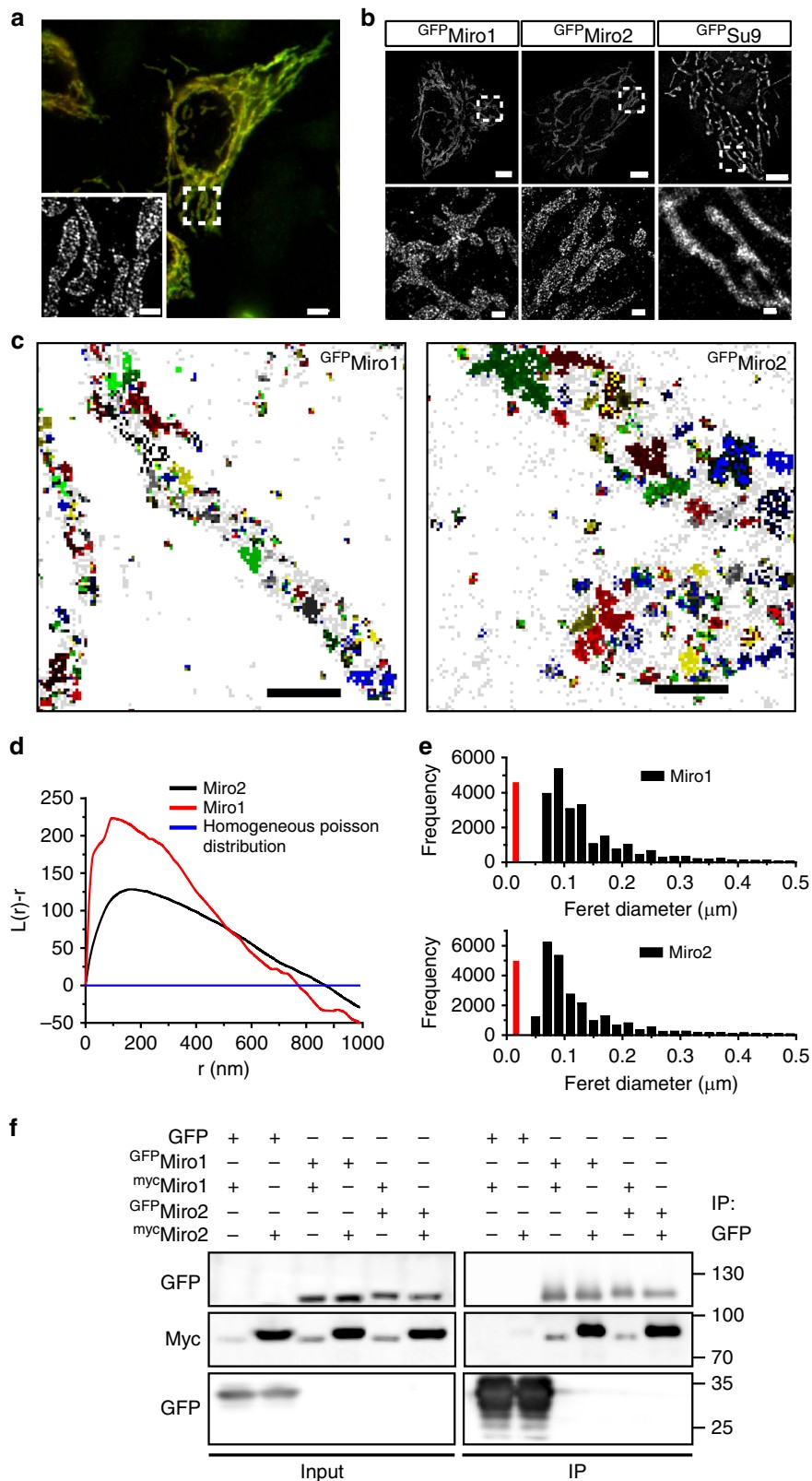

(NND)[43] between Miro2 and Mic60/Mitofilin, which showed that Miro clusters and Mic60/Mitofilin clusters are localized adjacent to each other in a periodic manner (Supplementary Fig. 8C). Similarly, <sup>GFP</sup>Miro1 clusters also positively correlated with Mic60/Mitofilin clusters (Supplementary Fig. 8D, E). In contrast, dual color dSTORM with <sup>GFP</sup>Miro2 and Tom20 (which we previously showed to be non-interacting with Miro proteins) showed

a significantly lower mean CCF peak when compared with <sup>GFP</sup>Miro2 and Mic60/Mitofilin (Fig. 5c–e) consistent with the specificity of the association between MICOS and Miro clusters.

**Miro1/2 regulates the MICOS complex formation and distribution.** Our data indicate that Miro proteins form clusters in

**Fig. 4 Sub-organellar localization of Miro proteins in the mitochondria. a** Widefield TIRF image of a representative HeLa cell overexpressing [myc]Miro1 and immunostained with anti-myc antibody. Inset shows the super-resolution image after structured illumination (SIM) (scale bar: 10 µm; inset: 1 µm). **b** dSTORM image of HeLa cells transfected with [GFP]Miro1, [GFP]Miro2, or [GFP]Su9. Both Miro1 and Miro2 localize to nanometer-sized clusters on the mitochondrial surface (scale bar: 5 µm; inset: 0.5 µm). **c** Density-based spatial clustering of applications with noise (DBSCAN) analysis of [GFP]Miro1 and [GFP]Miro2. Clustered localizations are represented by pseudo-color coding with localizations that are nonclustered is represented as gray pixels (scale bar: 0.5 µm). **d** Mean Ripley's K-function analysis of [GFP]Miro1 (Red) and [GFP]Miro2 (Black). Transformed K-function (L(r)-r) is represented against increasing cluster radius. The homogeneous Poisson distribution of the localizations is shown in blue ($n$ = cells; in which 13 Miro2 and 9 Miro1 cells from three independent experiments were used). **e** Size distribution of clusters formed by [GFP]Miro1 and [GFP]Miro2, respectively, in HeLa cells. After reconstruction of dSTORM images, mean Feret's diameter was measured using ImageJ and plotted. Red line represents localization precision of the dSTORM setup (median Feret's diameter; [GFP]Miro1 = 108 nm ± 85–162 nm and [GFP]Miro2 = 95 nm ± 67–150 nm; mean ± IQR; $n$ = 7 Miro2 cells comprising 23707 clusters and 7 Miro1 cells comprising 22876 clusters, from three independent measurements). **f** Western blot analysis of Miro1 and Miro2 interaction. HeLa cells were transfected with GFP as a control or [GFP]Miro 1/2 and [myc]Miro1/2. Immunoprecipitation was carried out using GFP-trap agarose beads and immunoblotted with GFP and myc antibodies

the mitochondrial surface that associate with MICOS clusters and interact with MICOS components and Sam50. Due to the alterations in cristae organization observed in Miro DKO cells, we aimed at understanding whether the loss of Miro proteins affects the interaction between the core components of the MIB/MICOS complexes spanning OMM and IMM. Both Mic19/CHCHD3 and Sam50 pulled down the core components of MIB/MICOS, e.g., Mic60/Mitofilin, Mic19/CHCHD3, and Sam50 (Fig. 5f). These interactions appeared conserved in the absence of Miro (Fig. 5f), indicating that there is no gross alteration of the core MICOS complex. PLA assays, which allows the in situ analysis of protein interactions, revealed that in absence of Miro there was a mild but significant decrease in the extent to which the core components of the MIB/MICOS interact (Fig. 5g, h). This weakening of inter-action was consistent between Sam50 and Mic60/Mitofilin, and between Mic60/Mitofilin and Mic19/CHCHD3 (Fig. 5g, h). Thus, while not essential for the assembly of MICOS complexes, Miro may regulate the overall stability of at least some species of these complexes, and its absence leads to the destabilization of parti-cular forms of the MICOS/MIB complexes.

Next, we wanted to directly test how MICOS organization at the IMM is affected by the loss of Miro. We carried out dSTORM imaging after staining WT and DKO cells against the core MICOS component Mic19/CHCHD3. In WT cells, Mic19/CHCHD3 showed an array of dense localizations evenly distributed throughout the entire mitochondria (Fig. 6a). In contrast, DKO cells showed mitochondrial regions with sporadic localizations of Mic19/CHCHD3 (Fig. 6a). In addition, DBSCAN analysis[40] in WT MEFs revealed the previously reported formation of Mic60/Mitofilin clusters arranged in a "discontin-uous rail-like distribution"[15] (Fig. 6a, b). Importantly, in DKO cells, this array of clusters was severely affected, with large areas of mitochondria devoid of Mic19/CHCHD3 clusters (Red circles in Fig. 6b). The heterogeneity of Mic19/CHCHD3 clusters distribution correlated with an altered NND distribution in DKO cells compared with WT (Fig. 6c). In comparison with WT MEFs, DKO cells showed decreased shorter NND distances (~60–110 nm), while longer NND distances (> 110 nm) were increased (Fig. 6c).

**Miro proteins link MICOS complex to the transport machin-ery.** Distribution of MICOS clusters is affected by the loss of Miro proteins. Due to its role in mitochondrial transport, we hypo-thesized that Miro might well serve as a link between MICOS clusters and the cytoskeleton. To test this hypothesis, we per-formed immunoprecipitation assays from WT and Miro DKO cell lysates with the core MICOS components and the two TRAK adaptor proteins. We observed a strong co-immunoprecipitation of TRAK1 with antibodies against Mic19/CHCHD3 or Sam50 (Fig. 6d). Strikingly, TRAK1 was no longer able to co-

immunoprecipitate with Mic19/CHCHD3 or Sam50 in Miro DKO cells, indicating that the interaction between TRAK1 and Mic19/CHCHD3 (and Sam50) is regulated by Miro (Fig. 6d). TRAK2 was also observed to co-immunoprecipitate with MICOS components only in WT cells although to a lower extent than that of TRAK1 (Fig. 6d), further supporting the Miro-dependent interaction between TRAK1/2 and MICOS. Reciprocally, both Sam50 and Mic19/CHCHD3 were readily detected in immuno-precipitates using a TRAK1 antibody from WT lysates, but not from DKO cell lysates (Fig. 6d). Thus, Miro proteins maintain an association between the cristae structures and the motor machi-neries through a complex containing MIB/MICOS components and the TRAK motor adaptor proteins.

We have recently shown that TRAK proteins can localize to, and induce the anterograde trafficking of, the mitochondria even in the absence of Miro[6]. The dependency on Miro of the TRAK1/MICOS association therefore suggested that the transport machinery regulated by Miro ensures that the pulling forces generated by the motors are directly applied to the MIB/MICOS complexes to facilitate the concerted transport of both mitochon-drial membranes. To test this, we transfected WT and DKO cells with a version of GFP fused to the first 70 amino acids of Tom70 to target the protein to the OMM (Tom70(1–70)[GFP])[44] and co-labeled endogenous Mic19/CHCHD3-positive clusters with specific antibodies. Again, these experiments showed that the Mic19/CHCHD3 signal distributed heterogeneously in DKO cells with some mitochondria showing high density of Mic19/CHCHD3 signal together with other mitochondria showing very low signal (Fig. 6e). To force the uneven transport of membrane compartments in our MEF models, we expressed TRAK1 and the motor KIF5C (together with Tom70(1–70)[GFP] to label mitochon-dria) and observed that mitochondria accumulated in the periphery of the cells both in WT and in DKO cells as expected[6]. Strikingly, in WT cells, the abundance of Mic19/CHCHD3 clusters matched the distal accumulation of the mitochondria, showing higher signal in distally transported mitochondria when compared with mitochondria that did not reach the periphery, thus suggesting that TRAK1/KIF5C-directed trafficking co-transported Mic19/CHCHD3-positive clusters (Fig. 6f). In contrast, in the absence of the TRAK1/MICOS bridge mediated by Miro, distally transported mitochondria were almost devoid of Mic19/CHCHD3 clusters in DKO cells (Fig. 6e, f), indicating that a critical role of Miro in regulating mitochondrial transport is to couple the TRAK/kinesin motor machineries to the MIB/MICOS complexes.

**TRAK/MICOS complexes control the distribution of IMM components.** By linking the mitochondrial transport machinery to MICOS clusters, Miro may coordinate the concerted transport of the OMM with the IMM containing the complexes responsible

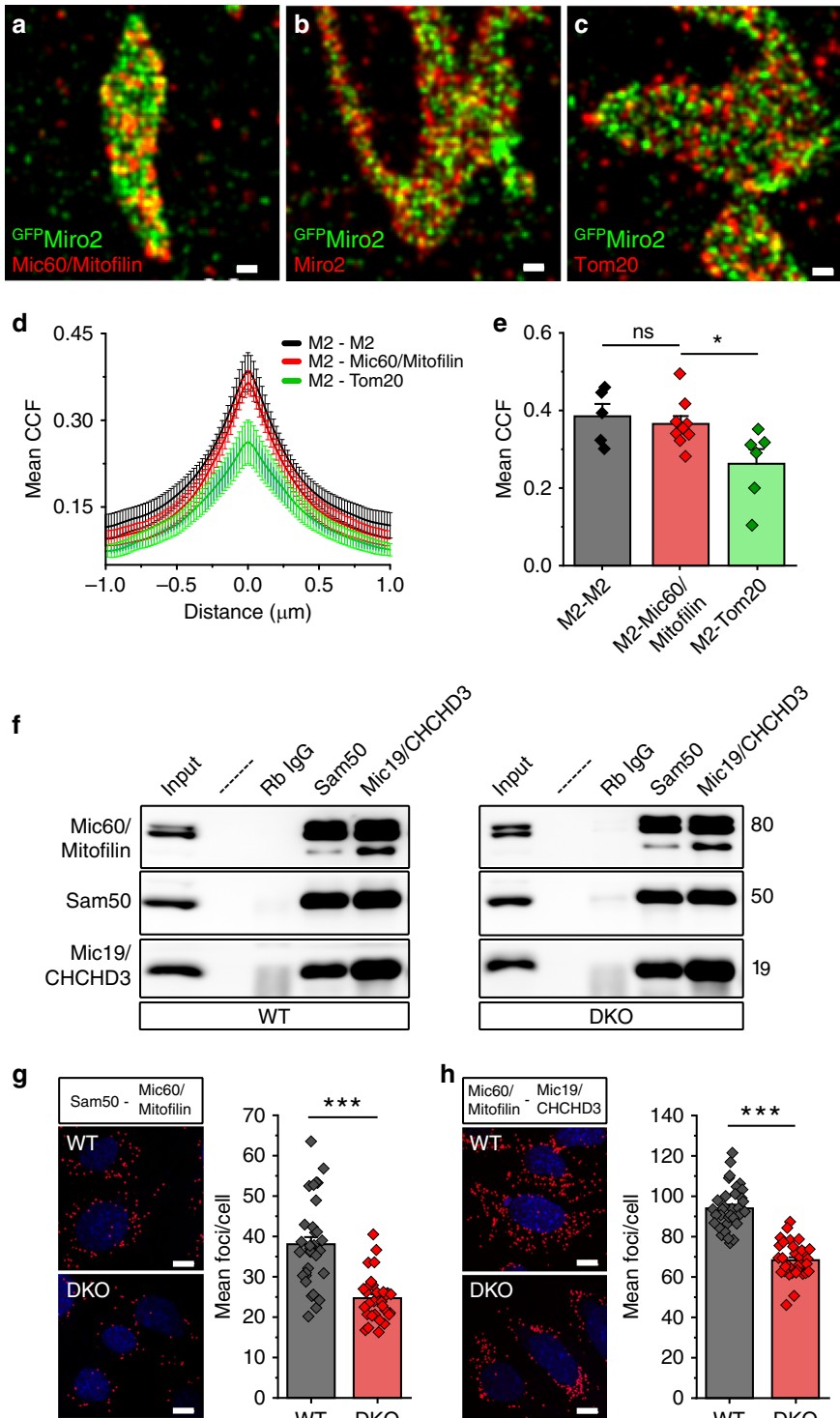

**Fig. 5** Miro nanodomains associate with MICOS clusters. **a–c** Dual color dSTORM imaging of [GFP]Miro2-transfected HeLa cells. [GFP]Miro2 nanometer-sized domains are shown (anti-GFP : green) together with endogenous Mic60/Mitofilin (**a**) a positive control for Miro2 (**b**; anti-Miro2 antibody) and negative control for Tom20 (**c**), (scale bar: 0.2 μm). **d** Cross-correlation analysis between Miro2 and Mic60; Miro2 and [GFP]Miro2 and Miro2 and Tom20 and **e** Mean Van Steensel's cross-correlation coefficient between [GFP]Miro2 and the different immunostainings calculated from (**d**) and plotted ($n =$ cells; in which 5 [GFP]Miro2-Miro2, 9 Miro2-Mic60/Mitofilin, and 6 Miro2-Tom20 cells from three independent measurements were used; One-way ANOVA, Bonferroni post hoc). **f** Endogenous immunoprecipitation experiment in WT and DKO cells using antibodies against the core-forming MICOS components (Mic19/CHCHD3, Mic60/Mitofilin and Sam50). The main interactions are not to be critically affected. **g, h** Proximity ligation assay (PLA) and quantification in WT and DKO cells between Sam50 and Mic60/Mitofilin (**g**) and between Mic60/Mitofilin and Mic19/CHCHD3 (**h**) show a mild decrease in the association between these components ($n =$ cells; in which 34 cells were used per condition from three independent experiments; Student's $t$ test with Welch's correction), (scale bar: 10 μm). Error bars represent ± SEM. Significance: $^*p < 0.05$; $^{***}p < 0.001$

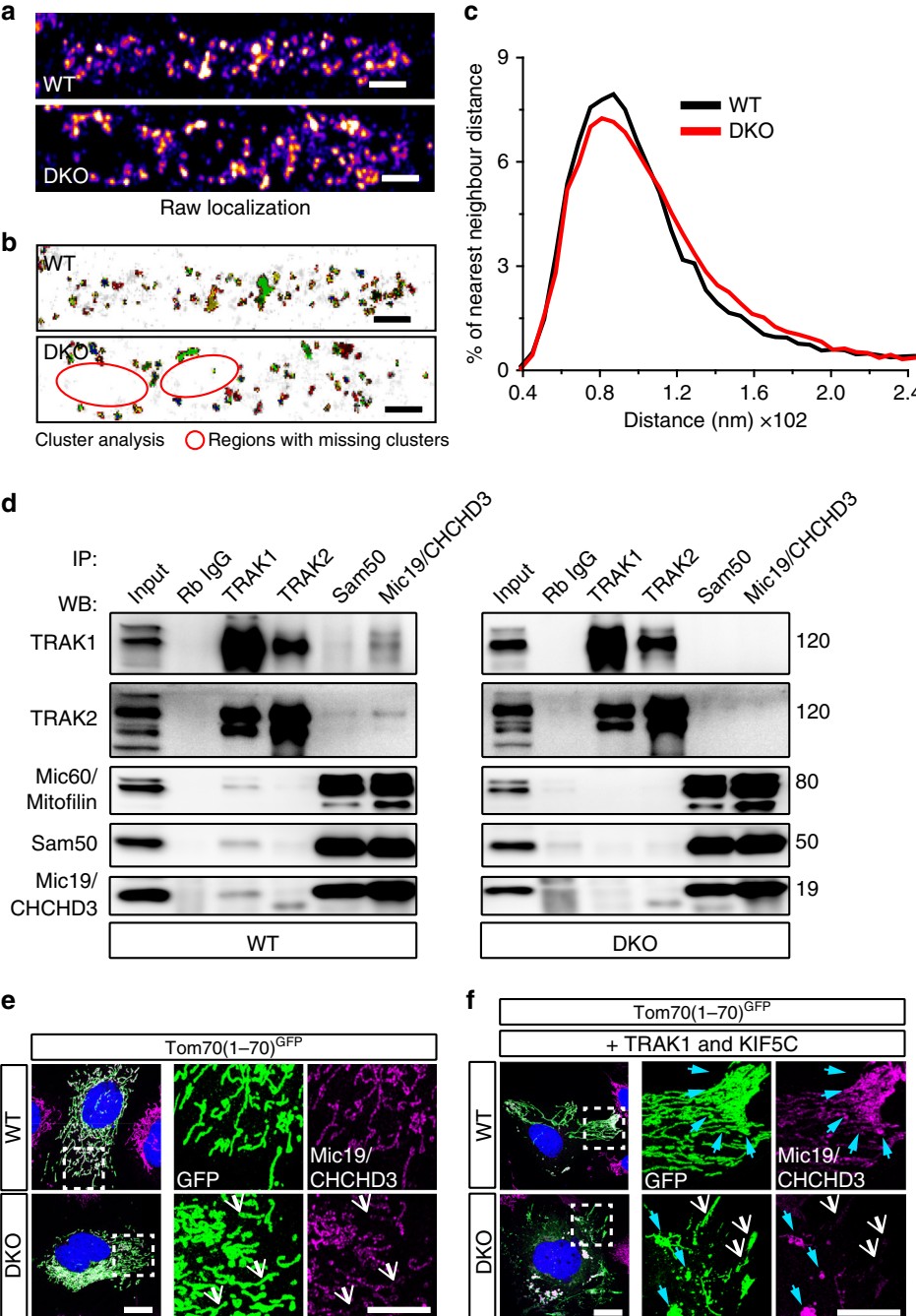

**Fig. 6** Altered distribution of MICOS in Miro DKO cells due to the loss of cytoskeletal anchorage. **a** Distribution of MICOS (Mic19/CHCHD3) clusters along the mitochondrial membrane in WT and DKO cells. Pseudo-colored representation of reconstructed dSTORM images of MICOS complexes (scale bar: 0.5 μm). **b** DBSCAN cluster map of Mic19/CHCHD3 clusters. DKO cells show the areas with depletion of clusters in comparison with WT cells (scale bar: 0.5 μm). **c** Quantification of nearest neighbor distances between MICOS clusters present in WT and DKO cells. Histogram of all the observed distances between MICOS clusters are shown. (WT: 8 cells, DKO: 11 cells comprising > 30,000 NND; $p < 0.001$, Kolmogorov–Smirnov's test). **d** Endogenous immunoprecipitation experiment in WT and DKO cells using antibodies against the core-forming MICOS components (Mic19/CHCHD3, Mic60/Mitofilin, and Sam50) and antibodies against the TRAK motor adaptors. TRAK proteins interact with critical MICOS components in a Miro-dependent manner. **e** Mic19/CHCHD3-positive MICOS clusters in WT distribute homogeneously in the mitochondrial population in WT MEFs, while the loss of Miro proteins correlates with an increase in heterogeneity in the distribution, with cells showing mitochondrial units almost devoid of Mic19/CHCHD3 signal (white arrows), (scale bar: 10 μm; insets: 5 μm). **f** Overexpressing TRAK1 and KIF5C induces the redistribution of the mitochondria to the periphery. In WT cells, this redistribution correlates with increased Mic19/CHCHD3 signal in the periphery (cyan arrows), (scale bar: 10 μm; insets: 5 μm). In DKO cells, TRAK1/KIF5C redistribution enhances the heterogeneity of Mic19/CHCHD3 staining, indicating a transport-mediated uncoupling of OMM and IMM in Miro DKO cells. Mitochondria with low Mic19/CHCHD3 signal concentrates in the TRAK1/KIF5C anterogradely transported mitochondria (white arrows), while Mic19/CHCHD3 signal accumulate in the proximal—not transported—mitochondria (cyan arrows)

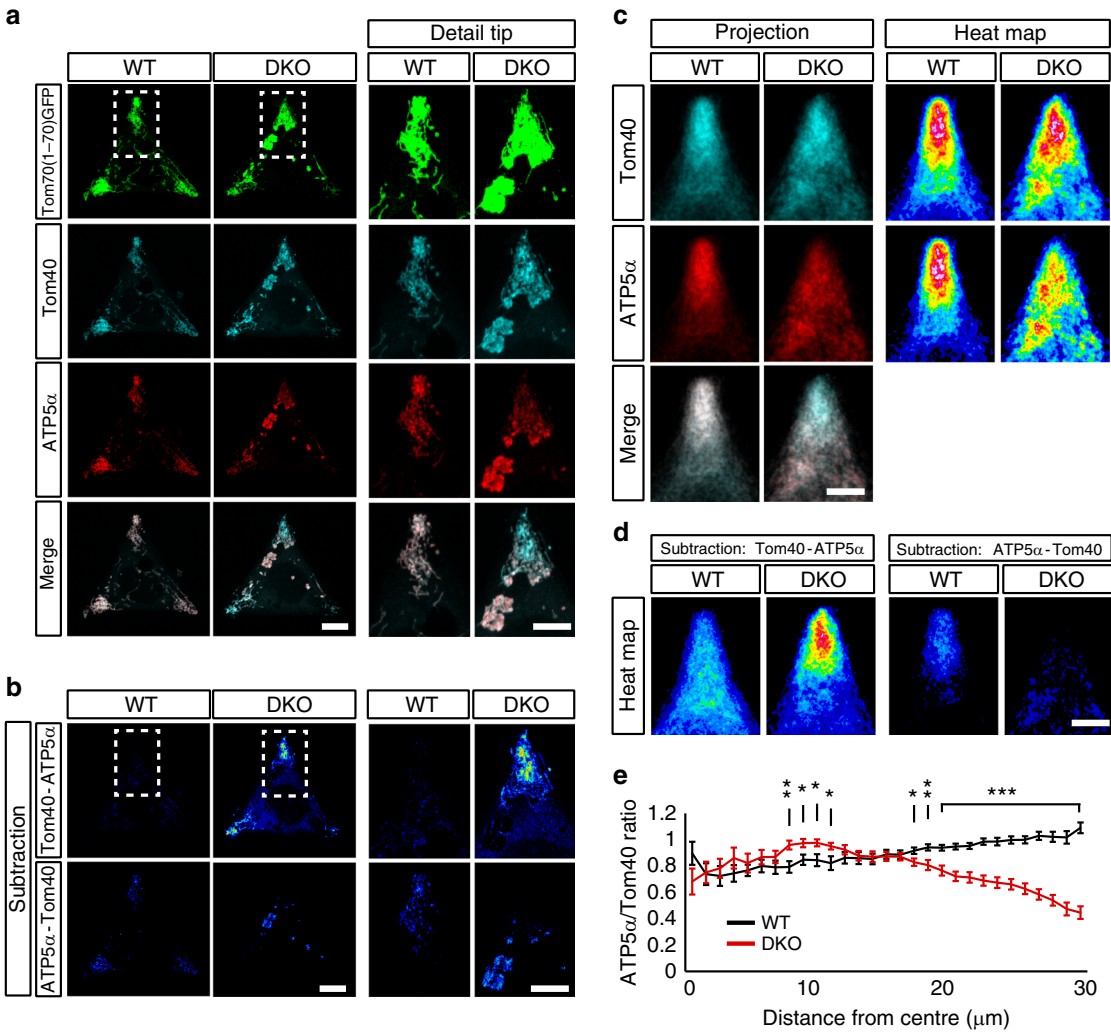

**Fig. 7** Miro links the microtubule transport pathway to the MICOS complex through TRAK. **a** Quantification of the distribution of OMM and IMM components upon TRAK1/KIF5C overexpression in micropatterned substrates. WT and MiroDKO cells expressing the Tom70(1–70)^GFP together with the mitochondrial motor machinery TRAK1/KIF5C were grown in "Y"-shaped micropatterns to produce triangular cells. Cells were immunostained for endogenous expression of an OMM marker (Tom40: cyan) and an IMM marker (ATP5α: red), (scale bar: 10 μm; insets: 5 μm). **b** Cell representations of the relative accumulations of the OMM marker Tom40 (subtraction of the ATP5α signal from the Tom40 signal; upper row) and the IMM marker ATP5α (subtraction of the Tom40 signal from the ATP5α signal, bottom row) (scale bar: 10 μm; insets: 5 μm). **c** Projections of all the 76 cell tips that contained the mitochondria and generation of mitochondrial probability maps. In WT cells, both OMM and IMM markers are similarly distributed, while in the Miro DKO cells the OMM marker is preferentially accumulated in the most distal regions compared with the IMM which accumulates in more proximal regions (scale bar: 5 μm). **d** Projections from the 76 subtracted images (for each genotype) were generated as in (**b**) (scale bar: 5 μm). **e** Mitochondrial probability map to quantify the ratio between the normalized signals of OMM (Tom40) and IMM (ATP5α) components as a function of the distance from the center of the cell (see Supplementary experimental procedures for details). All experiments were performed three independent times. Quantification and statistics in (**e**) were performed with 32 cells for each genotype (n = cells; Student's t test was performed at each distance point). Error bars represent ± SEM. Significance: *$p < 0.05$; **$p < 0.01$; and ***$p < 0.001$

for ATP generation. To test this hypothesis, we investigated the relative distribution of an IMM component of the OXPHOS system responsible for energy production (the ATPase subunit ATP5α) and an OMM protein, Tom40. We took advantage of our recently developed tools to accurately measure signal distribution in cells with restricted size and shape growing in adhesive micropatterned substrates (see Supplementary Methods for details)[6,45]. We again forced the redistribution of mitochondria to the periphery of the cells by expressing TRAK1 and KIF5C and measured the relative distribution of Tom40 and ATP5α on mitochondria (Fig. 7). In WT cells, both ATP5α and Tom40-signal presented a similar distribution in the mitochondrial network, consistent with a coordinated transport of both membranes (Fig. 7a, b). In stark contrast, DKO cells showed a relative

accumulation of the OMM marker, Tom40, in the periphery of the cells while the ATP5α signal appeared more accumulated in more proximal structures (Fig. 7a, b). Projection of all tips from all 32 cells imaged showed a consistent relative accumulation of the IMM marker (ATP5α) in the periphery of WT cells with respect to Tom40 signal, while in DKO cells the relative accumulation of the OMM marker in the periphery was accentuated (Fig. 7c, d). Furthermore, we also measured the density of signal in mitochondria of ATP5α and Tom40 in concentric rings radiating out from the center of the cell (MitoSholl analysis)[6,46] and calculated the ratios of the normalized signals (ATP5α/Tom40) to plot them as a function of distance (Fig. 7e). The resulting plot shows that, in WT cells, the ATP5α/Tom40 ratio increases toward the periphery indicating that Miro-regulated

TRAK1/KIF5C mitochondrial transport preferentially enriches the transported mitochondria with IMM components, perhaps by accumulating them by the pulling forces applied onto the MICOS complexes. In contrast, in DKO cells this ratio sharply drops in the most distal regions of the cell (Fig. 7e), indicating that, without Miro, TRAK-directed mitochondrial transport fails to efficiently couple the IMM to the mitochondrial transport pathway.

Altogether, these results suggest that Miro acts as a critical adaptor to link the mitochondrial transport machinery to the mitochondrial cristae organization to ensure the concerted transport of the OMM with the IMM components to guarantee the appropriate provision of energy to the regions where mitochondria are delivered.

## Discussion

Here, we demonstrate the nanoscale spatial organization and protein complex formation of the Miro mitochondrial GTPases and their dual role in regulating the formation and functionality of the ERMCS and in connecting the MIB/MICOS complexes, responsible for maintaining cristae architecture, to the mitochondrial transport pathway. Miro proteins link the TRAK motor adaptors to the MICOS complexes to ensure the correct distribution of MICOS throughout the mitochondria and to facilitate the coordinated delivery of both membranes during mitochondrial transport.

ERMCS are key structures for the regulation of $Ca^{2+}$ communication between the ER and mitochondria and play important roles in the regulation of mitochondrial division and the segregation of mitochondria and mtDNA in newly generated mitochondrial tips[47]. In yeast, the Miro homolog Gem1 is associated with the regulation of ER–mitochondria connections[8]. In mammals, Miro interacts with Mitofusins and DISC1, which are known to be associated with ERMCS[48–50]. Our data show that, in addition, the absence of Miro proteins leads to a decrease in contacts between the ER and mitochondria, which correlates with alterations in mitochondrial $Ca^{2+}$ uptake and in the intraluminal concentration of $Ca^{2+}$ in the ER. This role of Miro proteins in maintaining the ER–mitochondrial $Ca^{2+}$ homeostasis is supported by recent reports that link dMiro to the control of the VDAC1–IP3R complexes that regulate $Ca^{2+}$ communication in Drosophila[29,51]. Interestingly, increased levels of IP3R have been previously associated with increased ER $Ca^{2+}$ release, which in turn has an impact on muscle contractility, induction of apoptosis, and in the regulation of mitosis and that has been associated with multiple human diseases[52–54]. Altered $Ca^{2+}$ communication between the ER and mitochondria due to a decrease in ERMCS might be responsible for the increase in the protein levels of IP3R in Miro DKO cells. This provides striking evidence for the existence of a regulatory feedback mechanism that can control the number and composition of ER–mitochondrial contacts, depending on the activity of the ERMCS complexes. It is worth noting that the upregulation of IP3R might be a direct result of altered ER-associated degradation (ERAD) at ERMCS[55]. IP3R levels are controlled by ubiquitination, and the recent identification of the E3-ubiquitin ligase, Gp78, and other ERAD-associated proteins at ERMCS suggests that loss of ERMCS might affect the ubiquitination and subsequent proteasomal degradation of IP3R[51,56]. Our results suggest that this regulatory mechanism may be controlled by the levels or activity of Miro proteins although further studies are needed to uncover the molecular targets of Miro regulation.

Combined loss of both Miro proteins disrupts the architecture of mitochondrial cristae. This effect is reminiscent of the impact of depleting MICOS complex components, such as Mic60/Mitofilin, Mic19/CHCHD3, or some MICOS-associated proteins[14,57–60]. Knockdown of Sam50, an OMM protein, also results in the loss of cristae structure pointing at the key role of the MIB and cross-talk between inner and OMMs in regulating cristae architecture[18,32]. Our observations indicate a link between Miro and MIB/MICOS, which was previously postulated by both genetic and mass spectrometric based screens[19,34]. Super-resolution imaging shows that both Miro1 and Miro2 form nanoclusters of ~100 nm in size that are distributed throughout the mitochondrial network, reminiscent of clusters observed with MICOS proteins[15]. Moreover, dual-color dSTORM imaging indicates a close correlation in the distribution of Miro and MICOS protein clusters. Furthermore, we demonstrate that the loss of Miro proteins disrupts the previously reported "discontinuous rail-like" distribution of MICOS complexes throughout the mitochondria[15].

The disruption of mitochondrial cristae architecture by loss of Miro, while widespread, does not perfectly match that observed upon deletion of MICOS components or Sam50, which are usually described as "onion-like" membranous structures[60]. Instead, the majority of the mitochondria in Miro DKO cells presents large regions of the mitochondria with low density of cristae which often appear vesiculated. Because proteins embedded in the environment of the phospholipid bilayer are dynamic and can diffuse laterally[61], Miro proteins might provide structural support to the MIB/MICOS complexes through the TRAK adaptor interaction. MIB/MICOS complexes that have lost the Miro/TRAK anchor might be subject to uncontrolled lateral diffusion in the OMM leading to the loss of their "discontinuous rail-like" distribution[15]. In addition, it is possible that MIB/MICOS complexes that are not anchored to the cytoskeleton are rendered less stable and dissociate to a certain extent, explaining why we observe a small but significant decrease in the interaction between core components of the MIB/MICOS complexes in situ in our PLA assays.

Miro proteins are critical regulators of mitochondrial trafficking from yeast to mammals[5]. The accepted model of mitochondrial trafficking presumed that Miro proteins provided a link between the OMM and the microtubule motors kinesin and dynein through the recruitment of the adaptors TRAK1 and TRAK2[3,62–64]. We have recently challenged this idea by showing that Miro is not essential for kinesin/TRAK-directed mitochondrial movement, but rather regulates its activity[6]. In this paper, we demonstrate that an important function of Miro proteins is to regulate the association of the mitochondrial transport machinery to the MICOS complexes. By regulating this association, Miro proteins facilitate the concerted transport of both mitochondrial membranes to the cellular regions where they are needed. These findings are supported by a recent report linking Mic60/Mitofilin function, a core component of the MICOS complex, to mitochondrial motility in Drosophila[20]. Although the mechanism of such regulation remains unknown, in that work the authors report a decrease in dMiro levels upon genetic deletion of Mic60/Mitofilin. It remains to be tested if Miro is stabilized by the formation of assemblies between TRAK adaptors and MIB/MICOS complexes and whether the dissociation of these high-order assemblies renders Miro prone to be degraded.

A major consequence of this regulation is that the mitochondrial cristae that are associated to the OMM through the MIB complexes could potentially be distributed in a regulated manner by the concerted action of Miro and TRAK proteins. Thus, by connecting the transport machinery to the MICOS complexes, Miro opens the door to the concept of mitochondrial regulation by controlling the distribution of mitochondrial cristae within the mitochondria. In addition, the direct association with the cristae molecular architecture may provide a mechanism for the

transport machinery to sense the functionality of the mitochondria to be transported.

It has been known since the identification of the mitochondrial-associated membranes (MAMs) from the ER that the contact sites between both organelles are rich in enzymes necessary for phospholipid sysnthesis[65], and that ERMCS are important communication channels for the transport of lipids[66]. Therefore, together with the dissociation between MIB/MICOS complexes from the mitochondrial transport pathway, the decrease in ERMCS might have an impact in the lipid composition of the mitochondrial membranes and contribute to the altered mitochondrial ultrastructure observed in Miro DKO cells. The dual role of Miro in regulating the number of ERMCS and the distribution of MIB/MICOS complexes has parallels with the ER–mitochondria organizing network (ERMIONE) in *Saccharomyces cerevisiae*[67,68]. ERMIONE in yeast is formed by ERMES (ER–Mitochondria Encounter Structure) and MICOS which then recruit the TOM complex and Sam50 and is involved in lipid homeostasis, mitochondrial biogenesis, and maintenance of mitochondrial morphology[68]. However, there is little evidence of a structural molecular assembly in mammals that is homologous to the yeast ERMIONE. Our work supports a central role for Miro proteins in coordinating and integrating different mitochondrial functions by organizing and controlling a mitochondrial signaling network that includes the mitochondrial transport pathway, the MIB/MICOS complexes and the ERMCS and that might be the functional equivalent of the ERMIONE in mammalian cells.

## Methods

**Plasmid DNA, cell culture, and transfection.** $^{GFP}$Miro1 and $^{GFP}$Miro2 were generated as described earlier[69], while $^{myc}$Miro1, $^{myc}$Miro2[10,70], $^{myc}$Mfn1[71], and $^{GFP}$Su9 were obtained from Addgene. $^{GFP}$TOM70(1–70) was generated in the lab by cloning the first 70 residues of TOM70 into EGFP-N1 vector[44]. HeLa cells were purchased from ATCC. Mouse embryonic fibroblast were generated in the lab[6]. All cell lines were maintained in the DMEM medium (Gibco) supplemented with streptomycin (100 μg/ml), penicillin (100 U/ml), and 10% fetal bovine serum. Cells were transfected with ~5 μg plasmid DNA using nucleofection (Amaxa, Lonza AG) according to the manufacturer's protocol. Rat hippocampal cultures were prepared from E18 rat embryos[72,73]. Cells were seed at a density of 30,000–50,000 cells/cm$^2$ onto Poly-L-Lysine-coated coverslips and transfected at 7–9 DIV using Lipofectamine®2000.

All experimental procedures involving animals were carried out in accordance with institutional animal welfare guidelines and licensed by the UK Home Office in accordance with the Animals (Scientific Procedures) Act 1986.

**Immunostaining and biochemical assays.** Twenty-four to forty-eight hours post transfection, cells were fixed with 4% PFA at 37 °C for 10 min, washed, and coverslips were incubated for 30–45 min in blocking solution. Coverslips were then incubated with primary antibodies (see Supplementary Table 1 and Supplementary Methods for details) followed by secondary antibodies diluted in blocking solution and extensive washing to remove free fluorophores. Coverslips were further fixed for 8 min in 4% PFA, washed, and mounted for imaging.

For co-immunoprecipitation experiments in HeLa cells, cells were lysed 24 h post transfection in lysis buffer (50 mM Tris pH 7.5; 0.5% Triton; 150 mM NaCl; 1 mM EDTA; and 1 mM PMSF) containing protease inhibitors. The lysed samples were incubated with GFP-trap beads (Chromotek GmBH) for 1 h. Complexes were then washed several times and eluted with 3 × Laemmli sample buffer and western blotted on nitrocellulose membrane.

Co-immunoprecipitation experiments in MEF cells and in brains were performed similarly. Cells were collected and washed in PBS by centrifugation and adult brains were dissected prior to homogenization with lysis buffer (50 mM HEPES pH 7.5; 0.5% Triton; 150 mM NaCl; 1 mM EDTA; and 1 mM PMSF) containing protease inhibitors. Homogenates were cleared by centrifugation at 60,000×*g* for 40 min. One microgram of antibody was added to 1 ml of samples containing 2 mg of protein and incubated with rotation overnight at 4 °C. The next day, a mix 1:1 of ProtA and ProtG-coated agarose beads were blocked in lysis buffer containing 3 mg/ml of BSA for 1 h. After washing in lysis buffer, 20 μl of the beads mix was added to every tube and incubated for 1 h. Beads were then washed several times in lysis buffer and resuspended in Laemmli buffer, boiled for 5 min a kept at −20 °C until ran in acrylamide gels. Unprocessed scans of the western blots from the immunoprecipitation experiments in Fig. 3b and Fig. 6d are included in Supplementary Fig. 9.

Proximity ligation assay was performed with Duolink® In Situ Red PLA reagents according to the manufacturer's protocol (Sigma Aldrich)[49,74].

**Confocal, SIM, correlated SIM, dSTORM, and 3D dSTORM imaging.** Confocal imaging was performed on a Zeiss LSM 700 confocal microscope, Structured Illumination Microscopy was performed on Zeiss Elyra PS.1, correlated SIM, and dSTORM imaging was performed on the same microscope with 100 × 1.46 NA oil immersion objective. All dSTORM imaging was conducted using a custom-built microscope and analyzed using software written in C++ and Python[75]. Further details about super-resolution and electron microscopy performed in this study can be found in supplementary experimental procedures.

**Image processing and analysis.** Post reconstruction, images were first corrected for X-Y drift using one to three fiducials present in the images. Images were either binned using 20 -nm pixel size for dSTORM and colocalization with MICOS components. The reconstructed image was blurred with a Gaussian function with a sigma radius of 0.75 (which translate to 20–30 nm) using 'Accurate Gaussian blur' plugin. For measuring the sizes of nanoclusters, first images were thresholded, and then each particle was detected using particle analyzer algorithm followed by particle size measurement using Feret's diameter plugin present in ImageJ. For colocalization of dual-color STORM images, images in 555 -nm and 647 -nm channels were blurred equally then both channels were aligned using "Align images FFT" plugin present within GDSC ImageJ plugin (freely downloadable from University of Sussex) which uses a Gaussian for sub-pixel alignment. Van steensel's cross-correlation was calculated from the aligned images using plugin JACoP with X-shift of 1 μm. DBSCAN and Ripley's K-function were determined according to a previously published protocol[76].

**Statistical analysis.** Excel Software (Microsoft), Origin (OriginLab Corporation), and GraphPad Prism (GraphPad Software, Inc) were used to analyze the data. Statistical significance was calculated using two-tailed heteroscedastic Student's *t* test with Welch's correction for parametric data with unequal variance and Mann–Whitney *U* test for nonparametric data. Two-sample Kolmogorov–Smirnov test was used to compare the distributions of nearest neighbor distances of MICOS clusters. Statistical differences between multiple conditions were performed by one-way ANOVA followed by Bonferroni's post hoc tests. Normality of the data was assessed by applying one-sample Kolmogorov–Smirnov test. Statistical significance was pre-fixed at $P < 0.05$, described as *$p < 0.05$; **$p < 0.01$; and ***$p < 0.001$. All values in text are given as mean ± SEM unless specified.

**Reporting summary.** Further information on research design is available in the Nature Research Reporting Summary linked to this article.

## Data availability
All data supporting the findings of this study are available from the corresponding authors upon reasonable request.

## Code availability
All code used in this study is available from the corresponding authors upon reasonable request.

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

## Acknowledgements

This work was supported by an ERC starting grant (282430) and Lister Institute for Preventive Medicine prize to J.T.K. A.R.L. acknowledges support from the Medical Research Council award MR/K015826/1. S.M. was supported by an EMBO Long-Term Fellowship and Marie Skłodowska-Curie International Incoming Fellowship (Nos. 630033 and 913033). J.J.B. is supported by core funding to the MRC_UCL LMCB University Unit (Grant ref. MC_U12266B). E.F.H. received funding from the European Union's Horizon 2020 research and innovation programme under the Marie Skłodowska-Curie (No. 661733). D.I. was supported by an Institute of Neurology Clinical Neuroscience PhD and C.C.-C. by the MRC LMCB PhD program. J.T.K. and S.M. acknowledge support from the UCL super-resolution facility for the use of Zeiss Elyra PS.1 microscope and image analysis platform.

## Author contributions

S.M., G.L.-D., and J.T.K. conceived the project. S.M., G.L.-D., J.J.B., A.R.L., and J.T.K. designed experiments. S.M., G.L.-D., E.F.H., C.C.-C., D.I., D.M., and J.J.B. performed experiments and analyzed the data. I.L.A.-C. developed analytical tools. S.M., G.L.-D., and J.T.K. wrote the paper. G.L.-D. and J.T.K. supervised the project.

## Competing interests

The authors declare no competing interests.

## Additional information

**Peer review information** *Nature Communications* would like to thank Christian Eggeling and other, anonymous, reviewers for their contributions to the peer review information. Peer review reports are available.

