## [Peer Review File · Nature Communications]

Reviewers' comments:

Reviewer #1 (Remarks to the Author):

This manuscript by Kittler and colleagues describes an excellent series of high resolution microscopy studies of the mitochondrial Rho GTPase (Miro) in mammalian cells and tissues. They showed that Miro is localized as nanometer-sized particles on mitochondrial outer membrane. Based on biochemical and cell biological studies, the authors propose that Miro forms a complex with the mitochondrial outer membrane protein Sam50 and the MINOS complex at the mitochondrial inner membrane. They propose that this complex serves to strengthen the stability or integrity of mitochondrial cristae structure. They also propose that outer membrane localized Miro helps preserve the stability of ER-mitochondrial contact site (ERMCS). This study thus reveal new roles of Miro in mitochondrial physiology that are beyond the well characterized mitochondrial transport function of Miro. Overall, this is a well executed study and the topic is certainly of general interest to the readership of Nature Communications. There are a few issues that authors should consider addressing to improve the manuscript.

1. The Sam50 biochemical interaction data is weak. IP WB has background signals in IgG lane. Inclusion of additional controls such as the TOM complex proteins will test the specificity of Sam50-Miro interaction.
2. Functional role of Sam50 in linking Miro with MINOS should be strengthened using genetic knockouts or RNAi.
3. The cause of defects in cristae organization in Miro DKO remains unexplained. Is interaction between MINOS components or between MINOS and Sam50 weakened by Miro DKO?
4. The altered ER morphology in DKO also remains unexplained. Is IP3R-VDAC-MCU complex formation weakened by Miro DKO?

Reviewer #2 (Remarks to the Author):

The structural and functional organization of contact sites between mitochondria and the endoplasmic reticulum (ER) and their connections to key determinants of mitochondrial inner architecture have gained considerable attention in recent years. The term ER-mitochondria organizing network (ERMIONE) has been coined to describe the cooperating protein machineries that regulate inter-organelle crosstalk via contact sites and the formation of specialized membrane domains. The manuscript by Kittler and colleagues now implicates the Calcium-dependent Miro GTPases of mitochondria in these processes. Miro GTPases have been known previously for their role in the transport of mitochondrial particles within cells along cytoskeletal elements.

The authors now show that Miro1 and Miro2 assemble into protein clusters on the mitochondrial surface that partially co-localize with outer-inner mitochondrial membrane contact sites formed by the mitochondrial contact site and cristae organizing system (MICOS) and the SAM complex. Based on their biochemical experiments and their analyses of Miro1/2 DKO cells, they postulate that Miro proteins connect ER-mitochondria contacts sites to mitochondrial inner membrane architecture via MICOS.

This is a very interesting and appealing concept that reawakens the so far little noticed idea of Miro1 playing a crucial role at ER-mitochondria interfaces in mammals. This could be extrapolated from the reported interaction of Miro proteins with mitofusin-2 (Misko et al., 2010), likely a core component of ERMCS. The yeast Miro Gem1 has been shown to regulate the clustering of ERMES at organelle contact sites. Moreover, the Kittler lab recently reported on a connection between Miro proteins and

DISC1 (Norkett et al., 2016) that in turn has been shown to associate with MICOS (Park et al., 2010; Piñero-Martos et al., 2016). (Confusingly enough, none of these earlier papers is cited and discussed.)

Despite the exceedingly high general interest of the topic under investigation and the elegance of the proposed model, the experimental data do not sufficiently support the conclusions.

Major points:

1. To avoid confusion is it very important that the authors consistently(!) use the uniform consensus nomenclature for the mitochondrial contact site and cristae organizing system (MICOS) described in Pfanner et al. (2014). The name MINOS is outdated.
2. For the correct interpretation of the imaging data, it is critical to provide information on the spatial relationship of the observed Miro1/2 clusters with the endoplasmic reticulum.
3. Miro1/2 have been shown to interact with MFN1 and MFN2, which have been implicated in ER-mitochondria tethering. Does the clustering of Miro1/2 depend on MFN1 and/or MFN2? Does the partially co-localization of Miro1/2 with MICOS depend on MFN1 and/or MFN2? Western blots with antibodies against MFN1 and MFN2 shall be included in the co-IP experiments of Figure 4A.
4. Why do the authors conclude from the co-IP experiments in Figure 2g that Miro1/2 form heterodimers? The clusters observed in the imaging experiments are up to 200 nm in diameter, so there must be multiple copies of Miro1/2 in such a patch. The biochemical data cannot distinguish between Miro1/2 heterodimers and Miro multimers composed of Miro1 and Miro2 homo-dimers for example.
5. Figure 3 presents evidence for a partial co-localization of Miro clusters with MICOS (Mic60/mitofilin). As negative controls for proteins that do not co-cluster with Miro proteins, the authors use Tom20 and Mtx1. Especially the latter is very confusing, because numerous studies have shown that Mtx1 as well as other metaxins associate with the MICOS complex, likely via the contact site-forming Sam50 protein that makes up the human SAM complex together with at least Mtx1 and Mtx2. The MICOS-SAM/MTX interaction has been termed MIB complex. The Western Blot in Supplementary Figure 7 is not convincing, because input and elution of the IP are shown of separate gel/blot stripes and the eluate stripe is completely empty. There is no positive control for successful co-IP of any interacting protein in this experiment. Finally, the IP data in Figure 4A as well as the PLAs in Figure 4C show that Sam50 and Miro2 form complexes or are at least part of the same protein network, which does not fit with the absence of interaction between Miro proteins and Mtx1.
6. To test the proposed models of Miro/MICOS coupling the authors should test, if knock-down of Sam50 abolishes the co-localization and/or physical interaction between Miro and MICOS?
7. The IPs in Figure 4a and 4b lack any reasonable negative control demonstrating the specificity of the observed interactions. In Figure 4b also at least a Mic60/mitofilin western blot should be shown.
8. The BN-PAGE experiment in Figure 5c does not confirm to association of MICOS and Miro proteins in MEFs, because the Miro1 signal does not co-migrate with any of the indicated major MICOS bands. An original CHCHD3 western blot must be included in this figure.
9. A major problems of the manuscript is that the data presented in Figures 6 and S11 do not show a specific defect on cristae architecture in Miro1/2 DKO cells. The images simply shown deformed and damaged mitochondria, which might be targeted for mitophagy already. Is mitophagy increased or decreased in Miro1/2 DKO cells?

10. Related to this, the cristae morphology phenotype has no similarity with that of MICOS KO cells. Thus, the loss and deformation of cristae in Miro1/2 DKO cells cannot be caused by changes in MICOS. I do not understand how the data fit with the suggested connection between MICOS and Miro proteins (Figure 6 g)? The proposed changes in MICOS clustering in Miro1/2 DKO cells are not convincing, there seem to be no obvious differences. How exactly was the significance of the difference between the two curves in Figure 6i calculated. They seem to be virtually identical.

11. The quite dramatic changes in ER morphology are unlikely to be caused by the rather small reduction in ER-mitochondria contacts. Overall, the organelle morphology data showing massively altered morphologies of mitochondria and ER networks in Miro1/2 DKO cells rather raise doubts on the specificity of the apparent loss of ER-mitochondria contact sites.

Minor points:

The correlation with of Miro and Mic60 (mitofilin) clusters is a critical aspect of the paper, Therefore this correlation should also be shown for Miro1 in Figure 3 and Supplementary Figure 6.

If Miro1 and Miro2 form hetero-dimers (or hetero-oligomers), why do they run as almost completely separate bands of different sizes in BN-PAGE analysis (Figure 4B)? Weak Miro1 and Miro2 signals co-migrate with distinct forms (subcomplexes?) of the MICOS / MIB complexes in these gels. This also does not seem to fit with the heterodimer model for Miro1/2.

In order to judge the efficiency of interactions observed in the IPs shown in Figure 4a, the ratio of input and eluate should be given in %.

How do the authors interpret the up-regulation of IP3R and VDAC1 in Miro1/2 DKO cells? Is this a compensatory mechanism to counteract to potential loss of other (which?) type of ER-mitochondria contact site? From data in Figure 7 the authors conclude that the connecting interface between mitochondria and ER is rather decreased in these cells. How does that fit?

The ERMIONE concept should be mentioned in the discussion. Moreover the paragraph in the discussion on the possible cooperation of Miro, MICOS, Sam50 and OPA1 in cristae formation is highly confusing and speculative.

The idea that Miro proteins could fulfill a similar role in mammals like Mdm10 in yeast is difficult to reconcile with their totally different topology and structure. Mdm10 is a beta-barrel protein that is involved in mitochondrial protein import. The functional equivalents of key ERMES components in mammals are likely other TULIP domain-containing proteins, like the extended synaptotagmins.

Reviewer #3 (Remarks to the Author):

Muri et al present an advanced microscopy study on mitochondrial protein organization. Specifically, they investigate the spatial organization of Miro proteins with respect to clustering, the MINOS complex and ER-contact sites. The results highlight novel insights into the role of the Miro proteins. This study is very extensive, including various controls – it is a convincing, important and impressive story, which definitely deserves publication in Nature Communications.

I still would like to ask the authors to improve the description of their experiments and of their conclusions. The story is (as outlined) very extensive and therefore very long. It is sometimes hard to follow, but it does not take much to improve it. Here are some suggestions.

- The authors use different cell lines (HeLa, Mef), different microscopy approaches (SIM, dSTORM,

TEM, confocal, TIRF/non-TIRF) and analysis approaches (correlation, NND, ...). Although already done partially, it would be great to justify the use of each technique in each case – especially because they jump between the different cell lines (e.g. figure 1 uses Mef, the next figures HeLa) and experimental and analysis approaches. A simple 1-sentence statement each time would be sufficient.

- Have the cells always been fixed (has not been mentioned in the main text – I know it is in the methods section)? Also some further minor details could be mentioned in the main text such as how endogenous proteins have been labelled (it is sometimes said – just be consistent) or how GFP has been labelled to enable its use in dSTORM (to my knowledge it is not optimal to use it directly).
- I guess TIRF has been used for dSTORM – is the penetration depth good enough to image mitochondria?
- The authors include an impressive number of controls to ensure that what they see is real, such as clustering. They might consider a recent approach to exclude blinking artefacts (Baumgart et al Nat Methods 2016), and they should comment on corrections in the dual-color applications for drift, chromatic aberrations etc. Also, one might in one case compare different fixation methods (if not already done – did not become obvious to me).
- Page 7: Why has a control GFP-Miro1 and Myc-Miro2 and/or GFP-Miro2 and Myc-Miro2 not been included?
- Page 8: I disagree that the experimental results are a proof of dimers – certainly there are complexes, but these could be multimers. Where is the proof that these are exactly dimers?
- Sometimes, certain proteins are not well introduced such as Sam50 on page 10.
- Page 11, beginning: There is no direct proof that Sam50 is mediating the interactions – very speculative.
- Fig 4e: Maybe I mis-understood something. To me the faint band only proves that Miro is in large complexes; however these could also be non-associated with MINOS protein complexes.
- Page 12: I sometimes got confused whether wt or DKO cells were used – please specify each time (e.g. line 6).
- Page 12, line 7: Similarly to what?
- Page 12, middle: Why has the 150kDa band not been seen in figure 4e? What is the weight of Miro – state here as well?
- Page 12, bottom: Why has SIM been used here – any reason (see my comment above)?
- Figure 6c: This outcome should be described in more detail in the main text (as done for figure 6e).
- Figure 6e: Can the structures in the TEM images as well be rescued by re-expression of Miro?
- Page 14 top: Most major point – the authors claim that Miro proteins play key roles in the maintenance of cristae. However, the authors have only proven that it is important for general mitochondrial organization/phenotyping, but have not directly visualized cristae. Therefore, I consider this a slight overstatement.
- Page 14 middle: Why do the authors use confocal imaging here and not SIM or even dSTORM?
- Page 14 last lines: What are HRP constructs, what is ER retention, and what is the contrast of ER structures?
- Page 15 line 7: Two-times "this".
- Page 15 middle: The FRAP experiments could be explained in a better, more detailed way – hard to follow in the current version. The values of recovery times are given with too high precision (3 numbers after comma, although the errors are much higher).
- Page 16 discussions: Some of the statements (e.g. clusters as transport hubs or mitophagy initiators) seem very speculative – there is absolutely no proof for this (unless I missed something).
- Figure 1: Accurate label to y-axis in panel f is missing.
- Figure 4 panel c: What is red and what blue?
- Figure 5 panel c: Why is no data shown for CHCHD3 but only arrows?
- Figure 6: Why not just write SIM instead of Structured Illumination Microscopy? Panel c, labels need to be at the same level. Panel i: Any chance to plot a cumulative probability for other protein distributions as well, for comparison (so one can get a feeling for the extent of changes)?
- Figure 7 panels c-d: Mention what labels (m etc.) mean. Panel e: Is this a zoom-in from one of the images of panels c,d?
- Supplementary Information (SI) page 2 top: The type of antibody (mouse/rabbit) has not always

been mentioned.

- SI Fig. 1 caption line 1: Should this be "or" instead of "and"?
- SI Fig. 2, 3 captions: What z-stack (step size, range)?
- SI Fig. 4,12: What is green, what red?
- SI Fig. 5: Which cells?
- SI Fig. 6: Has the NND analysis been done on the confocal images of panels a.b?

Reviewer #4 (Remarks to the Author):

Mammalian Miro1 and Miro2 and their yeast homolog Gem1 are mitochondrial GTPases that are involved in various functions, including mitochondrial trafficking, formation of mitochondria/ER contact sites, and mitochondrial dynamics. Here, the authors analyzed localization, complex assembly, and function of mammalian Miro1 and Miro2. They propose that Miro proteins assemble into nanoclusters on the mitochondrial surface and interact via Sam50 with the MICOS complex in the mitochondrial inner membrane. Using knock out MEFs they report evidence that Miro proteins are important for cristae formation and ER contacts. The study reports some interesting observations, but it is in large parts preliminary.

Major points

1. A key conclusion of the authors is that Miro proteins in the outer membrane form super-complexes with MICOS in the inner membrane. Most of the experiments addressing this point were performed with over-expressed Miro. The authors show that endogenous Miro assembles into similar nanoclusters as overexpressed GFP-Miro, but they do not show co-localization of endogenous Miro with MICOS. More importantly, a co-immunoprecipitation of endogenous Miro with MICOS is lacking. Instead, the authors extensively examined an interaction of Miro with Sam50. However, Sam50 is not a "key component" of MICOS, as the authors incorrectly state in line 199. In sum, the experiments supporting complex formation of Miro with MICOS are rather preliminary.

2. The BN-PAGE experiments addressing complex assembly of Miro are not at all convincing. The bands highlighted with arrows in Fig. 4e are very weak or invisible. If Miro 1 and Miro2 are present in the same complex their band pattern should be very similar, which obviously is not the case in Fig 4e. Moreover, the WT BN-PAGE of Miro1 shown in Fig. 5c should be very similar to that in Fig. 4e, which also is not the case (also the arrows are not at the same positions).

3. The authors show that Miro DKO MEFs have reduced mitochondrial ER contacts. This will likely compromise the lipid composition of mitochondrial membranes and therefore affect cristae morphology. The differences in MICOS distribution observed in WT and DKO cells are rather subtle (Fig. 6) and could be simply due to the altered shape of the organelles. In sum, the data presented in the manuscript are not sufficient to reveal a direct role of Miro in cristae morphology, as it is stated in the title.

Further points

4. The authors should use the uniform nomenclature for the MICOS complex and its components (Pfanner et al. 2014; JCB 204:1083).

5. Line 64: The authors should add the citation of Stroud et al. 2011 (Journal of Molecular Biology), who identified an interaction of Gem1 with ERMES in parallel to Kornmann et al.

6. Is GFP-Su9 (mentioned in line 108) identical with Su9-EGFP (mentioned in line 419)? Please use a uniform nomenclature for the constructs. Also, the names and description of most plasmids are lacking. A simple reference to Addgene is not sufficient to unambiguously identify the plasmids. Please

cite the original papers.

7. I guess, GFP-Su9 shown in Fig. 1d is soluble GFP in the matrix. Why does it show punctate structures in dSTORM images?

8. The efficiency of CoIP is not clear, as it is not indicated how much input was loaded on the gels. The entire GFP blots should be shown (Fig. 2g and 4a). How many times were these experiments repeated?

9. Fig. 5a: Is "Miro" Miro1? Please explain the arrows in the legend.

10. Fig. 6e and f show quantifications of TEM data. It has to be stated somewhere how many cells were analyzed in how many independent experiments and whether the observed differences are statistically significant.

11. The results shown in Figs. S12 and S13 should be described in the Results section.

Reviewers' comments:

Reviewer #1

This manuscript by Kittler and colleagues describes an excellent series of high resolution microscopy studies of the mitochondrial Rho GTPase (Miro) in mammalian cells and tissues. They showed that Miro is localized as nanometer-sized particles on mitochondrial outer membrane. Based on biochemical and cell biological studies, the authors propose that Miro forms a complex with the mitochondrial outer membrane protein Sam50 and the MINOS complex at the mitochondrial inner membrane. They propose that this complex serves to strengthen the stability or integrity of mitochondrial cristae structure. They also propose that outer membrane localized Miro helps preserve the stability of ER-mitochondrial contact site (ERMCS). This study thus reveal new roles of Miro in mitochondrial physiology that are beyond the well characterized mitochondrial transport function of Miro. Overall, this is a well executed study and the topic is certainly of general interest to the readership of Nature Communications. There are a few issues that authors should consider addressing to improve the manuscript.

We thank this reviewer for finding this study interesting, well executed and of general interest to the readership of Nature Communications. We also thank this reviewer for their useful suggestions which we address point by point, below.

1. The Sam50 biochemical interaction data is weak. IP WB has background signals in IgG lane. Inclusion of additional controls such as the TOM complex proteins will test the specificity of Sam50-Miro interaction.

We have carried out additional experiments to strengthen the conclusion that Miro proteins interact with and regulate the function of the MICOS complex. We have expanded our co-immunoprecipitation experiments in HeLa cells to include other candidates related and unrelated to the MICOS complex. As suggested by this referee we have tested if a TOM complex protein, Tom20, co-immunoprecipitates with Miro proteins and we show that it does not (Fig. 3A of the revised manuscript).

To further support the endogenous Miro2/Sam50 interaction we have performed two different sets of experiments. First, we have used Proximity Ligation Assays (a technique that allows to test *in situ* the association of proteins that reside in close proximity (around 30 - 40 nm) within the same complex (Soderberg, Gullberg et al., 2006)). With these experiments we have confirmed close association between endogenous Miro2 and Sam50 and between Miro2 and Mic19/CHCHD3 (Fig. 3C). In addition, we also performed co-immunoprecipitation experiments in WT and Miro2^{KO} brains. In these experiments Miro2 antibody specifically co-immunoprecipitated both Sam50 and Mic60/Mitofilin in WT tissue but not in Miro2^{KO} brains (Fig. 3B). As positive controls for these experiments we report strong co-immunoprecipitation of Mic60/Mitofilin with Sam50 and Mic19/CHCHD3 antibodies. In addition, we also show that TRAK1, a known partner of Miro2, co-immunoprecipitates with Miro2 only in WT brains.

2. Functional role of Sam50 in linking Miro with MINOS should be strengthened using genetic knockouts or RNAi.

We thank the reviewer for this suggestion, however, in performing our endogenous Co-IP experiments in MEFs (both in WT and Miro DKO lines) we have observed that the interaction between the core components of the MICOS complexes (CHCHD3 and Mitofilin) with Sam50 are only mildly affected. This has led us to reconsider our interpretation and has implied the need for new experiments that we have designed and performed as follows:

In the revised manuscript, we further prove the specific interaction between Miro proteins and Sam50 as well as between Miro proteins and other members of the MICOS complex (Mic19/CHCHD3 and

Mic60/mitofilin in HeLa cells and Mic60/mitofilin in brain tissue; Fig. 3A, B). In addition, we have performed PLA assays to show that endogenous Miro2 can interact *in situ* with Sam50 and Mic19/CHCHD3 (Fig. 3C). Our data further support that endogenous Miro can interact with MICOS components and with Sam50. Furthermore, we are now providing evidence that Miro critically regulates the coupling of the MICOS/Sam50 complexes to the mitochondrial motor machinery through the mitochondrial transport adaptors TRAK1/2 (Fig. 6D). These Miro dependent assemblies are critical to couple the inner and the outer mitochondrial membranes to the mitochondrial transport machineries to maintain the spatial organization of the mitochondrial cristae (Fig. 6E). We think that Miro is interacting with the large complex formed by the association between Sam50 and the MICOS complex recently termed as Mitochondrial Intermembrane Bridge (MIB) or SAMMICOS (<http://dx.doi.org/10.1101/345959>) and links it to the transport machinery through TRAKs. In this way Miro regulates the concerted distribution of OMM and IMM ensuring the correct transport of the internal components required for energy production together with the outer mitochondrial membranes. We have modified this section and included 2 additional figures (Figs. 6 and 7) in the revised manuscript to support these conclusions.

3. The cause of defects in cristae organization in Miro DKO remains unexplained. Is interaction between MINOS components or between MINOS and Sam50 weakened by Miro DKO?

We agree with the reviewer that this is an important point that required clarification. We have improved the analysis of cristae morphology defects in Miro DKO cells to show that in the absence of Miro mitochondria shows a heterogeneous internal structure, discontinuous and often hollow when transfected with a matrix targeted GFP. We think that this effect is a consequence of the loss of higher order complexes in the absence of Miro, in which the TRAK adaptor and possibly other transport components are associated with MICOS components only if Miro is available. We provide functional evidence of our model by forcing the transport of mitochondria (through expression of TRAK1 and KIF5C) in the absence of Miro. Forcing the distal redistribution of mitochondria in Miro DKO cells leads to the relative accumulation of an IMM protein in proximal regions of the cell whereas OMM proteins get concentrated in distal regions. This indicates that the loss of Miro associates with an uncoupling between the inner and the outer mitochondrial membranes. We propose that this uncoupling is the reason why Miro DKO cristae are disorganised.

Further, to test whether the defects in cristae organization are due to a weakening of the interaction between MICOS components and Sam50 we performed co-immunoprecipitation studies. We have observed that there is no major disassembly between core MICOS components and Sam50 in MEF cells and in brain tissue (Fig. 3B and Fig. 5G). However, we observe a small but significant (20 - 25%) decrease *in situ* in the interaction between Sam50 and Mic60/Mitofilin or between Mic60/Mitofilin and Mic19/CHCHD3 as observed in our PLA assays (Fig. 3C). We propose that the loss of the high order associations between the MIB/MICOS complexes and the transport machinery would render the core MICOS complex less stable and this would explain the decrease in association between Sam50 and Mic60/Mitofilin or between Mic60/Mitofilin and Mic19/CHCHD3 that we report in our PLA assays.

4. The altered ER morphology in DKO also remains unexplained. Is IP3R-VDAC-MCU complex formation weakened by Miro DKO?

In the current version of the manuscript we show that a consequence of having a reduced level of ER/Mitochondria contact sites is the mishandling of Ca²⁺. The loss of Miro reduces the amount of calcium reaching the mitochondrial matrix upon ATP-induced calcium release from ER. This reduced Ca²⁺ uptake by mitochondria correlates with a higher amount of Ca²⁺ being released from the ER upon stimulation which would likely imply a higher cytoplasmic Ca²⁺ concentration. Interestingly, we also report a striking

increase in IP3R levels, a core component of the machinery that regulates Ca²⁺ release from the ER and that has been shown to be upregulated in conditions with severe defects in the communication between the ER and mitochondria (Takei, Mignery et al., 1994). We propose that the loss of Miro proteins may promote the IP3R-Grp75-VDAC axis as a compensatory mechanism to restore ERMCS and ER/Mitochondrial Ca²⁺ communication.

We also thank the reviewer for suggesting an alternative experiment by looking at IP3R-VDAC-MCU complex formation. It has been shown that in podocytes, an increased level of IP3R, Grp75, VDAC and MCU increases the interaction between IP3R, Grp75 and VDAC, and that this enhanced interaction in turn, drives calcium overload in mitochondria and induction of apoptosis (Xu, Guan et al., 2018). In addition, recent literature has shown that Miro directly interacts with MCU and controls calcium uptake (Niescier, Hong et al., 2018). Unfortunately, we haven't been able to consistently show interaction between the 4 proteins by immunoprecipitation [REDACTED]

[REDACTED]

Reviewer #2

The structural and functional organization of contact sites between mitochondria and the endoplasmic reticulum (ER) and their connections to key determinants of mitochondrial inner architecture have gained considerable attention in recent years. The term ER-mitochondria organizing network (ERMIONE) has been coined to describe the cooperating protein machineries that regulate inter-organelle crosstalk via contact sites and the formation of specialized membrane domains. The manuscript by Kittler and colleagues now implicates the Calcium-dependent Miro GTPases of mitochondria in these processes. Miro GTPases have been known previously for their role in the transport of mitochondrial particles within cells along cytoskeletal elements.

The authors now show that Miro1 and Miro2 assemble into protein clusters on the mitochondrial surface that partially co-localize with outer-inner mitochondrial membrane contact sites formed by the mitochondrial contact site and cristae organizing system (MICOS) and the SAM complex. Based on their biochemical experiments and their analyses of Miro1/2 DKO cells, they postulate that Miro proteins connect ER-mitochondria contacts sites to mitochondrial inner membrane architecture via MICOS.

This is a very interesting and appealing concept that reawakens the so far little noticed idea of Miro1 playing a crucial role at ER-mitochondria interfaces in mammals. This could be extrapolated from the reported interaction of Miro proteins with mitofusin-2 (Misko et al., 2010), likely a core component of ERMCS. The yeast Miro Gem1 has been shown to regulate the clustering of ERMES at organelle contact sites. Moreover, the Kittler lab recently reported on a connection between Miro proteins and DISC1 (Norkett et al., 2016) that in turn has been shown to associate with MICOS (Park et al., 2010; Piñero-Martos et al., 2016). (Confusingly enough, none of these earlier papers is cited and discussed.)

Despite the exceedingly high general interest of the topic under investigation and the elegance of the proposed model, the experimental data do not sufficiently support the conclusions.

We thank this reviewer for finding this model appealing and of high general interest. We have also taken their suggestion into account and included the above mentioned papers in the discussion section of the revised manuscript.

Major points:

1. To avoid confusion is it very important that the authors consistently(!) use the uniform consensus nomenclature for the mitochondrial contact site and cristae organizing system (MICOS) described in Pfanner et al. (2014). The name MINOS is outdated.

We thank this reviewer for their recommendations. We have updated the entire manuscript according to accepted nomenclature published in (Pfanner, van der Laan et al., 2014)

2. For the correct interpretation of the imaging data, it is critical to provide information on the spatial relationship of the observed Miro1/2 clusters with the endoplasmic reticulum.

We have addressed this issue by imaging ^{GFP}Miro1 or ^{GFP}Miro2 and labelling the ER with anti-PDI antibody. Cells were imaged under SIM to visualize Miro clusters and ER membranes, ER tubules were thresholded and a binary mask created. This mask was overlaid on to the mitochondrial image and clusters that are localized outside the ER are represented as magenta. The clusters that remain associated with ER are represented in cyan and accounted for 30-40% of total Miro clusters.

Reviewer's Figure 2. Localization of Miro clusters along ER tubules. HeLa cells transfected with ^{GFP}Miro1 or ^{GFP}Miro2, were immunostained with anti-GFP and anti-PDI antibodies and imaged under SIM. ER image (outline in red) was thresholded and masked. Fraction of ^{GFP}Miro1 that localizes in ER area: $34.2 \pm 5.4\%$ and ^{GFP}Miro2 that localizes in ER area: $44.3 \pm 3.6\%$ (Mean \pm SEM). The Miro clusters that are outside ER are presented in magenta and clusters overlapping with ER are shown in cyan. Scale bar 5 μm .

3. Miro1/2 have been shown to interact with MFN1 and MFN2, which have been implicated in ER-mitochondria tethering. Does the clustering of Miro1/2 depend on MFN1 and/or MFN2? Does the partially co-localization of Miro1/2 with MICOS depend on MFN1 and/or MFN2? Western blots with antibodies against MFN1 and MFN2 shall be included in the co-IP experiments of Figure 4A.

We have now performed and included the requested WB of Mfn1 and Mfn2 in the coimmunoprecipitation experiments. In contrast to the Misko study (Misko, Jiang et al., 2010) we were unable to detect any endogenous Mfn1 or Mfn2 being pulled down by Miro1 or Miro2 under our CoIP conditions. However, in the Misko study the authors used overexpressed Mitofusins which suggests that the interaction between Miro and mitofusins may be a relatively low affinity or a transient interaction, or be regulated by post-translational modifications.

In addition, we also tested whether overexpression of mitofusins might influence cluster size of Miro2 (see below). Our data shows that there is no significant difference in cluster size upon MFN1 overexpression, suggesting that Mitofusins do not influence clustering of Miro proteins. We have not included these experiments in the manuscript to simplify the message we want to deliver.

Reviewer's Figure 3. Clustering of Miro2 is independent of MFN1 overexpression. HeLa cells were transfected with GFP-Miro2 and Myc-MFN1, fixed and labeled with anti-GFP and anti-Myc antibodies. Dual-colour dSTORM imaging was performed as described earlier. Miro cluster size was measured as described earlier and presented as a box plot.

4. Why do the authors conclude from the co-IP experiments in Figure 2g that Miro1/2 form heterodimers? The clusters observed in the imaging experiments are up to 200 nm in diameter, so there must be multiple copies of Miro1/2 in such a patch. The biochemical data cannot distinguish between Miro1/2 heterodimers and Miro multimers composed of Miro1 and Miro2 homo-dimers for example.

We are sorry we have not been clear about this and agree with this reviewer that the section required further clarification. Our co-IP experiments allow us to conclude that Miro1 and Miro2 could be part of the same complex. From the BN-PAGE assays we observe the presence of a 150 kDa band that suggests that Miro1 and Miro2 could form dimers, and they could be both homo and/or heterodimers. We agree with the referee that multiple dimers can be part of higher MW complexes. We have now simplified our interpretation to avoid confusion.

5. Figure 3 presents evidence for a partial co-localization of Miro clusters with MICOS (Mic60/mitofilin). As negative controls for proteins that do not co-cluster with Miro proteins, the authors use Tom20 and Mtx1. Especially the latter is very confusing, because numerous studies have shown that Mtx1 as well as other metaxins associate with the MICOS complex, likely via the contact site-forming Sam50 protein that makes up the human SAM complex together with at least Mtx1 and Mtx2. The MICOS-SAM/MTX interaction has been termed MIB complex.

We agree with the reviewer that the use of Mtx1 as a negative control is confusing and we have modified this section for clarity. We now clarify that we use the OMM protein Tom20 as our negative control. Among the known MICOS components that we have considered we tested interaction between Miro proteins and Mtx1, which is known to interact with SAM50 to form the MIB (Huynen, Muhlmeister et al., 2016). We have not detected specific interaction between Mtx1 and either Miro1 or Miro2. This indicates that Miro proteins may form complexes with certain forms of MICOS but not necessarily all. We discuss the data in these terms in the revised version of the manuscript.

The Western Blot in Supplementary Figure 7 is not convincing, because input and elution of the IP are shown of separate gel/blot stripes and the eluate stripe is completely empty. There is no positive control for successful co-IP of any interacting protein in this experiment.

We understand this reviewer's concerns and we have addressed the issue by including in the same figure panel (Fig. 3A in the current version of the manuscript) all the co-IP assays. Each candidate was tested in at least 3 independent co-immunoprecipitations. We now show the western blot from inputs and elutions in the same image. In the panels there are now successful co-IPs between Miro proteins and several MICOS/MIB components (Sam50, Mic19/CHCHD3 and Mic60/Mitofilin). In the panel there are also proteins that did not coimmunoprecipitate demonstrating no interaction between Miro and other candidates (Tom20, Mfn1, Mfn2 and Mtx1). As a negative control we have used HeLa cells overexpressing GFP or the mitochondrially targeted ^{GFP} Su9 (Fig. 3A and Supplementary Fig. 2, respectively).

Finally, the IP data in Figure 4A as well as the PLAs in Figure 4C show that Sam50 and Miro2 form complexes or are at least part of the same protein network, which does not fit with the absence of interaction between Miro proteins and Mtx1.

We respectfully disagree with this interpretation. We believe that there are 2 possible explanations for this observation. First, it is possible that the potential interaction between Miro and Mtx1 in the same MIB/MICOS complex is indirect through one or more components in the same complex that are not strong enough to allow the pulldown of Mtx1 with anti-Miro antibodies. The other explanation is related to the composition of the different MIB/MICOS complexes. While there is wide consensus in ascribing central components to the core MICOS complexes there are different versions of these complexes that may carry different components. Metaxins (1 to 3) associate with Sam50 in high molecular weight complexes but not with Sam50 in lower molecular weight complexes. In fact, some studies have suggested that Mtx1 and Mtx2 may actually forms distinct complexes (Kozjak-Pavlovic, Ross et al., 2007) that are mutually exclusive of each other. In this scenario Miro proteins may associate with a subset of MIB/MICOS complexes that do not contain Mtx1. Because our co-immunoprecipitation and PLA experiments support an association between Miro and other MIB/MICOS components (like Mic19/CHCHD3 - Mic60/mitofilin or Sam50) for which we show a strong biochemical interaction, we believe that the later explanation is more consistent with our observations. Therefore, we propose that Miro proteins associates with certain forms of the MICOS complexes and associates them to the mitochondrial transport machinery through the TRAK adaptors.

6. To test the proposed models of Miro/MICOS coupling the authors should test, if knock-down of Sam50 abolishes the co-localization and/or physical interaction between Miro and MICOS?

We apologise for not having performed the suggested experiments. Briefly, (see point #2; referee #1) we have observed that the interaction between the core MICOS proteins and Sam50 is only mildly affected by the loss of Miro (Fig. 5H and I). This has led us to change our interpretation and perform additional experiments as follows. We now report that Miro rather links the Sam50/MICOS super-complexes to the mitochondrial transport pathway through the TRAK adaptor proteins. To consolidate this new view we have gathered different lines of evidence: We now further prove the specific interaction between Miro proteins and Sam50 and other members of the MICOS complex (Mic19/CHCHD3 and Mic60/Mitofilin) both in HeLa (Fig. 3A and C) cells and importantly, endogenously in brain tissue (Fig. 3B). Furthermore, we are now providing abundant evidence that Miro critically regulates the coupling of the MICOS/Sam50 complexes to the mitochondrial transport machinery through the mitochondrial transport adaptors TRAKs

(Fig. 6D). These Miro dependent assemblies are critical to couple the inner and the outer mitochondrial membranes to the transport machinery to maintain the spatial organization of the mitochondrial cristae (Fig. 6E, F and Fig. 7).

With these new data we think that Miro is interacting with the large complex formed by the association between Sam50 and the MICOS complex recently termed as Mitochondrial Intermembrane Bridge (MIB) or SAMMICOS (<http://dx.doi.org/10.1101/345959>) and links it to the transport machinery through the TRAK adaptors. In this way Miro regulates the concerted distribution of OMM and IMM ensuring the correct transport of the internal components required for energy production together with the outer mitochondrial membranes. We have modified this section and included 2 additional figures (Figs. 6 and 7) in the revised manuscript to support these conclusions.

7. The IPs in Figure 4a and 4b lack any reasonable negative control demonstrating the specificity of the observed interactions.

This is now addressed in Figure 3A in which we have included Tom20 as negative control.

In Figure 4b also at least a Mic60/mitofilin western blot should be shown.

We have followed this suggestion and we have now included Co-IP experiments from WT and Miro2KO mouse brains (Fig. 3B). In these experiments, western blots from the anti-Miro2 eluates show that endogenous Sam50 and Mic60/Mitofilin specifically co-immunoprecipitate with Miro2.

8. The BN-PAGE experiment in Figure 5c does not confirm to association of MICOS and Miro proteins in MEFs, because the Miro1 signal does not co-migrate with any of the indicated major MICOS bands. An original CHCHD3 western blot must be included in this figure.

We understand the reviewer's point and we have selected a better representation of the BN-PAGE western blot (Fig. 5F in the revised manuscript). In the new panel we show Miro immunoblotting (with the antibody that recognises both Miro1 and Miro2) together with Mic19/CHCHD3 from the same samples. Miro detection shows a band around 1200 kDa which is lost in Miro DKO MEFs and that matches with a band of the same molecular weight observed in blots with anti Mic19/CHCHD3 antibody. In addition Miro antibody also shows a band at around 550-600 kDa which is also present in Mic19/CHCHD3 blot. These two bands revealed with anti-Mic19/CHCHD3 are lost in Miro DKO cells lysates, suggesting that these two complexes in which Mic19/CHCHD3 is a constituent are destabilized in the absence of Miro. Furthermore, a new band appears in the Mic19/CHCHD3 western blot only in Miro DKO cells.

9. A major problem of the manuscript is that the data presented in Figures 6 and S11 do not show a specific defect on cristae architecture in Miro1/2 DKO cells. The images simply show deformed and damaged mitochondria, which might be targeted for mitophagy already. Is mitophagy increased or decreased in Miro1/2 DKO cells?

The role of Miro that we describe in the present manuscript specifically linking the organization of cristae junctions and their association with the mitochondrial transport machineries in a Miro dependent way is, we believe, self-containing. We have now included a more extensive description of the mitochondrial cristae observed in Miro DKO cells highlighting the vesiculation and the misdistribution of cristae throughout the mitochondrial space. We also provide evidence that Miro mediates the association of

MIB/MICOS complexes to the transport complexes which in turn would provide structural support for the cristae junctions. In the current version of the manuscript we extensively discuss these new data that we believe provides enough mechanistic insight to explain the observed phenotype of cristae morphology observed in Miro DKO cells.

However, we understand the point this reviewer makes regarding the possible implication of mitophagy in the observed phenotypes. We currently have a manuscript under consideration elsewhere in which we describe a role of Miro1 in regulating the induction and progression of the mitophagic process which we have pre-published in bioRxiv (bioRxiv 414664; doi: <https://doi.org/10.1101/414664>). Briefly, we find that the loss of Miro resulted in the delayed recruitment of Parkin to mitochondria in response to mitochondrial damage. In addition we show that in a later step of the mitophagic process Miro degradation is required to allow for appropriate mitochondrial clearance. This data is in line with a recent report that also describes a role for Miro1 in the initial recruitment of Parkin to mitochondria. However, the impact on mitophagy is observed upon the loss of Miro1 alone (Safiulina, Kuum et al., 2019), which does not lead to dramatic remodelling of cristae, suggesting that these observations are independent of an effect on mitophagy.

10. Related to this, the cristae morphology phenotype has no similarity with that of MICOS KO cells. Thus, the loss and deformation of cristae in Miro1/2 DKO cells cannot be caused by changes in MICOS.

As per the previous point, we have now intended to provide a deeper description of the cristae morphologies observed in our system. We have also provided a wider representation of the mitochondrial internal structure in Figure 1A, C and E. We propose a model by which Miro proteins control the formation of high order complexes linking the MIB/MICOS complexes with the transport pathway through the TRAK proteins. Furthermore, we also report that some interactions between MICOS components are relatively weakened *in situ*, as shown by our PLA assays (Fig. 5H and I), possibly as a consequence of having lost the attachment provided by the TRAK adaptors. Thus, the cristae morphologies induced by the missing anchorage between MICOS and the transport machinery would be observed as a compound phenotype of altered cristae morphologies together with defects in distribution of the mitochondrial cristae junctions. We believe that the current description is not at odds with prior knowledge in the field as, in our interpretation, the core components of the MIB/MICOS complexes would still be functional in linking IMM and OMM although the higher organization of these cristae and its distribution throughout the mitochondria would be highly affected by the missing connection between MICOS and the transport machinery. We have extensively discussed this point in the current discussion.

10.b. I do not understand how the data fit with the suggested connection between MICOS and Miro proteins (Figure 6 g)? The proposed changes in MICOS clustering in Miro1/2 DKO cells are not convincing, there seem to be no obvious differences. How exactly was the significance of the difference between the two curves in Figure 6i calculated. They seem to be virtually identical.

We agree with the referee that the images provided were not clear enough due to limits in the resolution provided, although in our opinion, the data and analysis performed in the former Fig. 6B was still valid. The analysis calculated the difference between the distributions of nearest neighbour distance (Significance was calculated using two sample non-parametric hypothesis using both Mann-Whitney U and Kolmogov-Smirnov D tests, both of them produced significant p-values: $p < 0.005$). To address the issue we have repeated the experiment using STORM). Under STORM the distribution of clusters positive for Mic19/CHCHD3 in WT vs Miro DKO mitochondria can be easily observed and compared between conditions. The new images are now part of Figure 6 (Fig. 6A). Next, we performed cluster analysis of the images and Density-based spatial clustering of applications with noise (DBSCAN) based cluster maps on

WT and Miro DKO cells. When we compared DBSCAN maps we observed less periodicity of Mic19/CHCHD3 clusters in Miro DKO cells compared to WT cells. The analysis of raw localizations and cluster formation from these localizations revealed more frequent absence of clusters in Miro DKO cells (red circles in Fig. 6B). We have quantified this effect calculating the Nearest Neighbour Distances (NND) which is now in panel Figure 6C. However, in order to better represent the differences we have chosen to represent the data as a distribution curve (as opposed to the cumulative distribution in the previous version) in which, we believe, it is easier to interpret the data. In short, the clusters positive for Mic19/CHCHD3 distribute in a more uniform fashion in WT mitochondria while in Miro DKO mitochondria these clusters distribute more heterogeneously. We have included a brief explanation in these lines when introducing these data in the revised manuscript.

11. The quite dramatic changes in ER morphology are unlikely to be caused by the rather small reduction in ER-mitochondria contacts. Overall, the organelle morphology data showing massively altered morphologies of mitochondria and ER networks in Miro1/2 DKO cells rather raise doubts on the specificity of the apparent loss of ER-mitochondria contact sites.

In the revised version of the manuscript we show that as a consequence of having a reduced level of ER-mitochondrial contacts there is an important dysregulation of the ER/mitochondrial Ca^{2+} communication as seen by the decreased mitochondrial Ca^{2+} uptake after ATP induced Ca^{2+} release from the ER. This reduced Ca^{2+} uptake by mitochondria correlates with a higher amount of Ca^{2+} being released from the ER which would likely imply a higher cytoplasmic Ca^{2+} concentration. In addition we also report a 4-fold increase in the levels of IP3R which has been already shown to be upregulated in conditions with severe defects in the communication between the ER and mitochondria and that has also been proposed to be responsible for altered ER morphology in cultured cells and neurons (Takei et al., 1994).

Minor points:

The correlation with of Miro and Mic60 (mitofilin) clusters is a critical aspect of the paper, Therefore this correlation should also be shown for Miro1 in Figure 3 and Supplementary Figure 6.

We thank the reviewer for raising this point. We have now added the data showing positive correlation between Miro1 and Mic60/Mitofilin clusters. This new data can be found in Supplementary figure 8.

If Miro1 and Miro2 form hetero-dimers (or hetero-oligomers), why do they run as almost completely separate bands of different sizes in BN-PAGE analysis (Figure 4B)? Weak Miro1 and Miro2 signals co-migrate with distinct forms (sub-complexes?) of the MICOS / MIB complexes in these gels. This also does not seem to fit with the heterodimer model for Miro1/2.

We acknowledge that the strongest bands in the BN-PAGE gels from HeLa lysates (Fig. 3F) indicate that the majority of Miro1 and Miro2 are not in the same complexes. However, the bands observed at around 140 KDa are strongly indicative that they form dimers (Fig. 3G). Together with the immunoprecipitation experiments (Fig. 4A) the data demonstrates that these dimers might be both homodimers and heterodimers. How these dimers are organized in high molecular weight complexes is less clear but the fact that some bands around 1000 - 1200 KDa or 600 - 700 KDa can be seen with both Miro1 and Miro2 antibodies from HeLa cells lysates supports that Miro1/Miro2 could be part of the same high molecular weight complexes. We have clarified this in the results section when the data is introduced.

In order to judge the efficiency of interactions observed in the IPs shown in Figure 4a, the ratio of input and eluate should be given in %.

This is mentioned now in figure legend (Fig. 3 in the current version of the manuscript) and in Experimental procedures. Approximately 2% of total cell lysate used for immunoprecipitation was loaded for inputs.

How do the authors interpret the up-regulation of IP3R and VDAC1 in Miro1/2 DKO cells? Is this a compensatory mechanism to counteract to potential loss of other (which?) type of ER-mitochondria contact site? From data in Figure 7 the authors conclude that the connecting interface between mitochondria and ER is rather decreased in these cells. How does that fit?

As the referee points out we interpret the increase in IP3R levels as a compensatory mechanism to counteract the loss of ER/Mitochondrial contacts and to maintain the ER-Mitochondria Ca^{2+} communication. We propose that Miro proteins regulate ER/Mitochondrial calcium communication by controlling the number of ERMCS. Loss of both Miro proteins associates with a decrease in ERMCS which would impact the Ca^{2+} uptake inside mitochondria upon agonist-induced Ca^{2+} release from ER. In agreement with this we observe both the amplitude and the rise time of Ca^{2+} peak is significantly decreased in Miro DKO mitochondria compared to WT (Fig. 2H and I). IP3R is a core component of the machinery that regulates calcium release from the ER. It has been shown to be upregulated in disease conditions with alterations in the ER-associated degradation (ERAD) pathway that control levels of IP3R and that show defects in Ca^{2+} communication between the ER and mitochondria (Abou-Saleh, Pathan et al., 2013, Pearce, Wang et al., 2007).

The ERMIONE concept should be mentioned in the discussion. Moreover the paragraph in the discussion on the possible cooperation of Miro, MICOS, Sam50 and OPA1 in cristae formation is highly confusing and speculative.

We have now replaced this section and added a discussion about the ERMIONE complex in the revised manuscript.

The idea that Miro proteins could fulfill a similar role in mammals like Mdm10 in yeast is difficult to reconcile with their totally different topology and structure. Mdm10 is a beta-barrel protein that is involved in mitochondrial protein import. The functional equivalents of key ERMES components in mammals are likely other TULIP domain-containing proteins, like the extended synaptotagmins.

We agree with this reviewer and we have now removed the sentence in the revised manuscript.

Reviewer #3

Muri et al present an advanced microscopy study on mitochondrial protein organization. Specifically, they investigate the spatial organization of Miro proteins with respect to clustering, the MINOS complex and ER-contact sites. The results highlight novel insights into the role of the Miro proteins. This study is very extensive, including various controls – it is a convincing, important and impressive story, which definitely deserves publication in Nature Communications.

I still would like to ask the authors to improve the description of their experiments and of their conclusions. The story is (as outlined) very extensive and therefore very long. It is sometimes hard to follow, but it does not take much to improve it. Here are some suggestions.

We thank this reviewer for finding our work impressive, extensive and their recommendation for publication. We also thank them for the valuable suggestions which we have addressed below.

- The authors use different cell lines (HeLa, Mef), different microscopy approaches (SIM, dSTORM, TEM, confocal, TIRF/non-TIRF) and analysis approaches (correlation, NND, ...). Although already done partially, it would be great to justify the use of each technique in each case – especially because they jump between the different cell lines (e.g. figure 1 uses Mef, the next figures HeLa) and experimental and analysis approaches. A simple 1-sentence statement each time would be sufficient.

Following this recommendation we have now included a brief description and justification of cell lines and techniques used throughout the manuscript.

- Have the cells always been fixed (has not been mentioned in the main text – I know it is in the methods section)?

Now each figure legend has the information about the imaging condition (e.g. fixed or live cell imaging). Actually, only a subset of the imaging was done with fixed cells. Mostly we used PFA based fixation for dSTORM/SIM imaging while deconvolution microscopy and FRAP were performed under live imaging conditions.

Also some further minor details could be mentioned in the main text such as how endogenous proteins have been labelled (it is sometimes said – just be consistent) or how GFP has been labelled to enable its use in dSTORM (to my knowledge it is not optimal to use it directly).

It is now mentioned throughout the main text. We have also provided a Supplementary Table 1 listing all the antibodies used in each figure. GFP was either labeled with anti-GFP antibody (used in Fig. 4C and D, Supplementary Fig. 3) followed by secondary antibodies conjugated to Alexa 647 or anti-GFP nanobody (Used in Fig. 5A and Supplementary Fig. 6) that was labeled with Alexa 647 in house.

- I guess TIRF has been used for dSTORM – is the penetration depth good enough to image mitochondria?

We indeed carried out dSTORM in imaging buffer (90% Glycerol in Tris pH 8.0+5% VectaShield H1000) in which TIRF mode provided the best signal to noise (S/N) ratio. The improved S/N further enabled accurate localization precision. Our approach provided a depth of field of about 1.5 μm and within these volumes most of the mitochondrial network is represented.

- The authors include an impressive number of controls to ensure that what they see is real, such as clustering. They might consider a recent approach to exclude blinking artefacts (Baumgart et al Nat Methods 2016),

We thank the reviewer for suggesting further analysis to confirm that the clustering by Miro proteins observed is not arising due to multiple blinking events from the same antibody. In applying the suggested analysis (Baumgart, Arnold et al., 2016) we could not achieve Varying Label Density essential in performing the analysis. However, we have performed additional cluster analysis using three published methods that help in assessing the blinking artefacts often observed associated with dSTORM: (i) we performed cluster analysis by pair wise correlation methods as recently described (Sengupta, Jovanovic-Taliman et al., 2011). We chose a region with mitochondria and a region with background (with a similar number of localizations) and plotted the autocorrelation function ($G(r)$) over distances (r). As a control we also simulated a random distribution with a similar number of localizations as observed for GFP^{Miro2} . According to Sengupta et al. a random distribution of molecules will show a single exponential decay while molecules showing clustered distribution will show decay mostly fitted by a two exponential model. When we tried fitting the $G(r)$ over distances (r), the GFP^{Miro2} showed bi-exponential fitting while background showed single exponential fitting. This suggest that GFP^{Miro2} might exist as clusters along mitochondrial membrane (Reviewer's Figure 4A). In an alternative approach, we also measured Ripley's K function (Owen, Rentero et al., 2010) in both GFP^{Miro1} and GFP^{Miro2} expressing cells and observed peak clustering is highest around 100 nm and extend until 800 nm. Both of these analysis further support that GFP^{Miro1} and GFP^{Miro2} exist as clusters along mitochondrial membrane (Reviewer's Figure 4B). Also, we implemented DBSCAN analysis (Malkusch & Heilemann, 2016) and the cluster map of both GFP^{Miro1} and GFP^{Miro2} showed heterogenous domains of Miro with various shapes and sizes along mitochondrial outer membrane (Reviewer's Figure 4C and D).

Reviewer's Figure 4: Cluster analysis of GFP tagged proteins. (A) Pair autocorrelation plot and (B) Ripley's K function plot over distance. Both of the analysis indicates strongly the clustered domains of Miro along outer mitochondrial membrane. (C and D) DBSCAN cluster map showing spatial organization of ^{GFP}Miro1 ^{GFP}Miro2 respectively. Scale bar: 500 nm.

and they should comment on corrections in the dual-color applications for drift, chromatic aberrations etc.

A detailed section on correction is mentioned in the experimental section of the supplementary information. We performed the drift correction as described in (Lowe, Tang et al., 2015). Briefly, we used fluorescent microspheres (100 nm) as fiducials to correct for drifts and real-time focus locking performed during the imaging as well as post-imaging translational drift correction.

Also, one might in one case compare different fixation methods (if not already done – did not become obvious to me).

We mostly performed 4% PFA (with 5% sucrose containing 1mM Ca²⁺ and 1mM Mg²⁺) fixation method for 5-10 min at room temperature. This is now mentioned in the figure legends of the corresponding sections. We have tried other fixation methods, such as methanol fixation followed by immunofluorescence but lead to fragmentation of mitochondria. Therefore, we decided to continue our studies using a PFA fixation method.

- Page 7: Why has a control GFP-Miro1 and Myc-Miro2 and/or GFP-Miro2 and Myc-Miro2 not been included?

We chose conditions by keeping in mind the best possible expressing constructs. ^{Myc}Miro2 expression is not adequate to perform a dSTORM experiment. The labeling did not provide enough blinking events to reconstruct a representative image for quantification.

- Page 8: I disagree that the experimental results are a proof of dimers – certainly there are complexes, but these could be multimers. Where is the proof that these are exactly dimers?

We understand the reviewer's point and have modified this section accordingly. We agree with this referee that Fig. 4A can only confirm interaction and it is now mentioned (also please see rev.2 point 4 for similar concerns). Now we conclude from Fig. 4A (Formerly Fig. 2) that Miro1 and Miro2 might be part of the same complexes and that they may form dimers between them (both homo- and heterodimers) which is supported in Fig. 3G, where a band is detected with both antibodies at 140 kDa approx. (which is 2x MW of individual Miro protein) only in WT cells. Our experiments do not address whether Miro proteins form higher molecular complexes made up of dimers, monomers or even multimers.

- Sometimes, certain proteins are not well introduced such as Sam50 on page 10.

We have now addressed this and provided a clear introduction about all the proteins including Sam50.

- Page 11, beginning: There is no direct proof that Sam50 is mediating the interactions – very speculative.

We have performed more co-immunoprecipitation experiments in HeLa cells (Fig. 3A) and, endogenously, in brain lysates (Fig. 3B). We show that endogenously, Miro can interact with both Sam50 and MICOS components. In addition, we also carried out proximity ligations assays (PLA) to confirm the *in situ* interaction between Miro2 and Sam50 and between Miro2 and Mic19/CHCHD3 (Fig. 3C). Our data suggest that endogenous Miro can interact with components of the MIB/MICOS complex. Since we cannot conclusively demonstrate that Sam50 links Miro to MICOS we have removed the sentence suggesting that the Miro-MICOS interaction is possibly through Sam50.

Moreover, in our revised version of the manuscript we now show that both Mic19/CHCHD3 and Sam50 interact with TRAK proteins in a Miro dependent way (Fig. 6D). We propose that Miro proteins regulate both the correct distribution of cristae junctions through this interaction as well as the concerted transport of both mitochondrial membranes to where mitochondria are delivered.

- Fig 4e: Maybe I mis-understood something. To me the faint band only proves that Miro is in large complexes; however these could also be non-associated with MINOS protein complexes.

We agree with the referee in his statement. However, in addition to co-migrating in similar molecular weight complexes on BN-PAGE membranes we now show that Miro proteins interact with different components of the MIB/MICOS complexes. Furthermore, we show that certain high molecular complexes that contain Mic19/CHCHD3 are lost while new, lower molecular weight bands, appear instead. Altogether, this further supports that Miro is part of high order complexes related with MICOS.

- Page 12: I sometimes got confused whether wt or DKO cells were used – please specify each time (e.g. line 6).

We apologize for not mentioning this clearly in the main text. This has now been corrected and each figure clearly mentions the cell type used throughout the revised manuscript.

- Page 12, line 7: Similarly to what?

We are sorry for the lack of clarity. The ^{GFP}Miro2 staining that was mentioned in the main text was similar to the staining observed in HeLa cells and cultured neurons. This suggests that Miro cluster formation is independent of cell type. This is now more clearly explained.

- Page 12, middle: Why has the 150kDa band not been seen in figure 4e? What is the weight of Miro – state here as well?

In Figure 4e, the blot was cropped to focus on the bands that migrate to a similar position as MICOS complex proteins. Miro proteins are present around 140 kDa in those blots as well. For clarity, a full blot (including the 140 kDa band corresponding to Miro) from both HeLa and MEFs cells is presented below.

Reviewer's Figure 5. BN-PAGE of Miro proteins. BN-PAGE was performed on whole cell extracts from WT or DKO cells. The dimeric Miro proteins bands appeared at ~140 kDa. The asterisk represents non-specific bands recognized by Miro antibodies.

- Page 12, bottom: Why has SIM been used here – any reason (see my comment above)?

Due to resolution limit of conventional confocal microscopy we decided to use SIM microscopy, which has successfully been used to image mitochondrial matrix (Fiolka, Shao et al., 2012), to analyse mitochondrial matrix continuity in our WT and Miro DKO cell lines.

- Figure 6c: This outcome should be described in more detail in the main text (as done for figure 6e).

This section has now been expanded and described in more detail. This data has been moved to Figure 1 in the revised manuscript.

- Figure 6e: Can the structures in the TEM images as well be rescued by re-expression of Miro?

We are sorry we have not performed the EM experiments rescuing with Miro1 or Miro2 expression in Miro DKO cells. To achieve this we would need to perform Correlative Light-Electron Microscopy (CLEM) to be able to image cells which we can confirm are expressing our constructs. Because our experiments are based on scoring populations of transfected cells it would have required, in our opinion, an excessive amount of work. Although lower resolution than TEM, SIM imaging allows the visualization of a matrix targeted GFP with enough resolution to identify whether expression of Miro1 or Miro2 were able to rescue the vesiculated and often hollow mitochondrial matrix of Miro DKO cells. We believe that the rescue in

the continuity of the mitochondrial matrix that we report allows us to suggest that the cristae morphology phenotype observed by EM would similarly be rescued.

- Page 14 top: Most major point – the authors claim that Miro proteins play key roles in the maintenance of cristae. However, the authors have only proven that it is important for general mitochondrial organization/phenotyping, but have not directly visualized cristae. Therefore, I consider this a slight overstatement.

We have now addressed the cristae phenotype in more detail in the revised manuscript. We have selected more representative images of mitochondrial cristae from WT and DKO cells (Fig. 1C and E) to show that in the absence of Miro mitochondria shows a heterogeneous internal structure, discontinuous and often hollow when transfected with a matrix targeted GFP.

- Page 14 middle: Why do the authors use confocal imaging here and not SIM or even dSTORM?

We quantified the ER/mitochondria overlap using conventional microscopy methods as this approach has been used by several research groups for analysis of ERMCS previously (de Brito & Scorrano, 2008, Harmon, Larkman et al., 2017, Horner, Liu et al., 2011). Spinning disk confocal microscopy allowed us to perform live-cell imaging of large stacks in very short periods of time. Live imaging is critical to study ER structures as fixation procedures are known to generate artefacts in tubular ER structures. In addition, the speed of acquisition allowed the imaging of large number of cells used to produce our dataset which included 4 different genotypes. Other imaging methods like SIM or dSTORM would have required the fixation of the sample (as ER and mitochondria are highly dynamic organelles) and the throughput would not have allowed the study on the same amount of cells which we believe is critical for this analysis.

- Page 14 last lines: What are HRP constructs, what is ER retention, and what is the contrast of ER structures?

We transfected WT and DKO cells with an HRP (Horseradish Peroxidase) tagged with an ER retention signal (KDEL) to specifically label ER (Connolly, Futter et al., 1994). In the presence of H₂O₂, the HRP substrate, DAB (3, 3'-diaminobenzidine) is oxidized resulting in a product that precipitates in the lumen of the ER and can be visualized by both light and electron microscopy. After the DAB oxidation we treated the cells with osmium tetroxide which forms an insoluble electron-opaque precipitate that can be observed under EM. Due to this osmium tetroxide precipitate ER appeared as electron dense structures in Fig. 2E. This is now mentioned in the figure legend (Fig. 2) and in the experimental section.

- Page 15 line 7: Two-times "this".

This is now corrected

- Page 15 middle: The FRAP experiments could be explained in a better, more detailed way – hard to follow in the current version. The values of recovery times are given with too high precision (3 numbers after comma, although the errors are much higher).

We have now discussed this section in more detail and have reduced the precision of the values to 2 decimals. The fluorescence of ER-luminal targeted DsRed (^{DsRed}ER) in Miro DKO cells recovered similar as WT after photobleaching suggesting that overall connectivity between the tubules remained unchanged.

- Page 16 discussions: Some of the statements (e.g. clusters as transport hubs or mitophagy initiators) seem very speculative – there is absolutely no proof for this (unless I missed something).

In light of our new data regarding the formation of high order complexes between MIB/MICOS and TRAK adaptors in a Miro dependent manner we believe we provide enough support for proposing that the Miro clusters associated with MICOS components are molecular platforms for the anchorage of the motor machinery through the TRAK adaptors that regulate both mitochondrial transport and the distribution and arrangement of cristae junctions in the mitochondria. We, however, agree with the referee regarding the idea of the clusters having a role as mitophagy initiators is speculative and have removed this from the discussion.

- Figure 1: Accurate label to y-axis in panel f is missing.

We have now removed this panel and only kept the histogram of Feret's diameter.

- Figure 4 panel c: What is red and what blue?

Fig. 4C: Nucleus is stained using DAPI and represented as Blue, while PLA assay products (red) contains a Texas Red conjugated complementary oligonucleotide that binds to amplified DNA upon rolling circle amplification. An increased punctate signal is indicative of close proximity between two proteins (<30 nm). This has now been specified in the figure legend of Fig. 3C.

- Figure 5 panel c: Why is no data shown for CHCHD3 but only arrows?

We have carried out additional experiments with Miro DKO cells and now we present a BN-PAGE with antibodies against Mic19/CHCHD3 for better comparisons. Please see new Fig. 5F.

- Figure 6: Why not just write SIM instead of Structured Illumination Microscopy? Panel c, labels need to be at the same level.

We have addressed this issue and replaced Structured Illumination Microscopy with SIM everywhere but for the first time mentioned. We apologize for the mis-alignment of panel C. Please see Fig. 1B (Formerly Fig. 6C) for corrected representation.

Panel i: Any chance to plot a cumulative probability for other protein distributions as well, for comparison (so one can get a feeling for the extent of changes)?

We are sorry we have not included an additional cumulative distribution plot for other proteins. STORM analysis is highly dependent on the imaging conditions and must be carried at the same time, within the same experiments. In the current manuscript, however, we have replaced the cumulative distribution plot

of Nearest Neighbour Distances (NND) with the histograms (Fig. 6C). We believe that the histogram of NND better represents that the Mic19/CHCHD3 clusters distribution in WT mitochondria is more homogeneous than in Miro DKO mitochondria. Additionally, we have also performed DBSCAN cluster map of the MICOS complex along mitochondria. In WT MEFs the distribution of Mic19/CHCHD3 clusters is more homogeneous and resembles the “discontinuous rail like distribution” recently described (Jans, Wurm et al., 2013) while in DKO mitochondria this arrangement is significantly affected with more missing clusters (Fig. 6A - B).

- Figure 7 panels c-d: Mention what labels (m etc.) mean. Panel e: Is this a zoom-in from one of the images of panels c,d?

We have now changed the labels for coloured arrows or asterisks which are explained in the figure legends. Regarding Fig. 7E (now Fig. 2E in the current version of the manuscript), the image is indeed a zoom of the WT image. The mitochondria at the bottom right corner is zoomed to show close contacts between ER and mitochondria. This is now clearly mentioned in the figure legend.

- Supplementary Information (SI) page 2 top: The type of antibody (mouse/rabbit) has not always been mentioned.

We have addressed this and the species of each antibody are now mentioned. Additionally, we have also provided a table in the supplementary information detailing all the antibodies used in each figure panel.

- SI Fig. 1 caption line 1: Should this be "or" instead of "and"?

We thank the reviewer for highlighting this. However, this figure has now been removed from current supporting info.

- SI Fig. 2, 3 captions: What z-stack (step size, range)?

Step size and range is now mentioned supporting experimental procedures. In SIM, we typically acquire stacks with 0.14 μm and about 50 planes corresponding to $\sim 7\text{-}8 \mu\text{m}$. We typically cover the whole cell in this range but in some cases the cells were slightly thicker in Z which required up to 10 μm range. Supplementary Fig. 2 and 3 are now Supplementary Fig. 3 and 4 respectively.

- SI Fig. 4,12: What is green, what red?

Supplementary Fig. 12 have been removed from the current manuscript for clarity. In Supplementary Fig. 4 (now Supplementary Fig. 5), anti-Myc antibodies (used to detect ^{Myc}Miro1) were labelled with the same secondary antibody labelled with two different fluorocroms (Alexa-488 and Alexa-647). The green image represents the detection of Alexa-488 under SIM while red represents dSTORM image (of the same field) acquired in the Alexa-647 channel. This is now clear in the figure.

- SI Fig. 5: Which cells?

This is now Supplementary Fig. 6. These images were taken after expressing ^{GFP}Miro2 construct in HeLa cells. This is now mentioned in the corresponding figure legend.

- SI Fig. 6: Has the NND analysis been done on the confocal images of panels a.b?

This is now Supplementary Fig. 8). All the NND analysis were performed using dual-color dSTORM. Specifically, ^{GFP}Miro2-Mic60/Mitofilin NND is measured from images quantified in Fig. 5A (formerly Fig.3A and E).

Reviewer #4:

Mammalian Miro1 and Miro2 and their yeast homolog Gem1 are mitochondrial GTPases that are involved in various functions, including mitochondrial trafficking, formation of mitochondria/ER contact sites, and mitochondrial dynamics. Here, the authors analyzed localization, complex assembly, and function of mammalian Miro1 and Miro2. They propose that Miro proteins assemble into nanoclusters on the mitochondrial surface and interact via Sam50 with the MICOS complex in the mitochondrial inner membrane. Using knock out MEFs they report evidence that Miro proteins are important for cristae formation and ER contacts. The study reports some interesting observations, but it is in large parts preliminary.

Major points

1. A key conclusion of the authors is that Miro proteins in the outer membrane form super-complexes with MICOS in the inner membrane. Most of the experiments addressing this point were performed with over-expressed Miro. The authors show that endogenous Miro assembles into similar nanoclusters as overexpressed GFP-Miro, but they do not show co-localization of endogenous Miro with MICOS.

We are now providing strong evidence that Miro proteins endogenously interact with MICOS components in MEF cells and in brain lysates. We agree with the referee that we don't show colocalization of endogenous Miro with MICOS clusters, however, we would like to point out that successful cluster analysis and correlation between clusters is highly dependent on the specificity of the antibody and a high signal to noise ratio. We show that endogenous Miro2 localizes in clusters in the mitochondrial membrane (Supplementary Fig. 4) visualised under SIM. Unfortunately, neither Miro1 nor Miro2 specific antibodies has allowed us to perform STORM imaging as the staining is not strong enough to provide adequate blinking in order to reconstruct dSTORM images. We believe that our imaging experiments together with the biochemical and functional approaches, provide enough support to conclude that Miro proteins form clusters that co-localises with MICOS clusters.

More importantly, a co-immunoprecipitation of endogenous Miro with MICOS is lacking. Instead, the authors extensively examined an interaction of Miro with Sam50. However, Sam50 is not a "key component" of MICOS, as the authors incorrectly state in line 199. In sum, the experiments supporting complex formation of Miro with MICOS are rather preliminary

We agree with the referee regarding nomenclature used in line 199 from the previous manuscript. We have now corrected this issue and we specifically talk about MICOS or Mitochondrial Intermembrane space Bridging complex (MIB) whenever necessary.

Regarding the complex formation between Miro and MIB/MICOS complexes we have now widened our biochemical experiments to provide supporting evidence for a role of Miro proteins in regulating the MICOS complexes. We now show that Miro2 interacts endogenously with Sam50 and with Mic60/Mitofilin as they are pulled-down with specific antibodies against Miro2 from WT brain lysates but not from Miro2 KO brains (Fig. 3B). We have also performed Proximity Ligation Assays (PLA), also called *in cell* co-immunoprecipitation, to show that, *in situ*, Miro2 is in close proximity (around 30 - 40 nm (Soderberg et al., 2006)) to both Sam50 and a MICOS component, CHCHD3 (Fig. 3C). Finally, we now also show the endogenous interaction between the motor adaptors, TRAK1 and TRAK2, with critical components of the MIB/MICOS complexes CHCHD3, mitofilin and Sam50 from MEF lysates. Importantly, these endogenous interactions are Miro dependent as they fail to occur in Miro DKO MEFs (Fig. 6D).

2. The BN-PAGE experiments addressing complex assembly of Miro are not at all convincing. The bands highlighted with arrows in Fig. 4e are very weak or invisible. If Miro 1 and Miro2 are present in the same complex their band pattern should be very similar, which obviously is not the case in Fig 4e. Moreover, the WT BN-PAGE of Miro1 shown in Fig. 5c should be very similar to that in Fig. 4e, which also is not the case (also the arrows are not at the same positions).

We agree with the referee that our representation was not clear enough and have addressed the issue. Some of the differences in the banding pattern that the referee points out are due to differences in the cells used. The gels in Fig. 3F and G (Former Fig. 4c) are from HeLa cell lysates and show a different banding pattern than MEFs cells (Fig. 5F). BN-PAGE gels from MEFs lysates from WT and DKO cells allows the identification of Miro specific bands, including complexes around 500-700 and about 1000/1200 kDa that are present in WT and which disappear in Miro DKO cells (Fig. 5F). We have also included side by side with the Miro western blots, the banding pattern of a MICOS component, Mic19/CHCHD3, in WT and Miro DKO cells in the same figure (Fig. 5F). The changes in the Mic19/CHCHD3 banding pattern in the absence of Miro indicates that Miro regulates the assembly of some of the complexes positive for Mic19/CHCHD3.

3. The authors show that Miro DKO MEFs have reduced mitochondrial ER contacts. This will likely compromise the lipid composition of mitochondrial membranes and therefore affect cristae morphology. The differences in MICOS distribution observed in WT and DKO cells are rather subtle (Fig. 6) and could be simply due to the altered shape of the organelles. In sum, the data presented in the manuscript are not sufficient to reveal a direct role of Miro in cristae morphology, as it is stated in the title.

We understand this reviewer's concerns, however, studies in yeast knocked out for Gem1 have shown that, while affecting ER-mitochondrial communication by altering the organization of ERMES, there is no major disruption of lipid transfer between the organelles (Kornmann, Osman et al., 2011, Nguyen, Lewandowska et al., 2012). Therefore, we believe that the cristae deformation that we report in our work is due to the alteration in MICOS assembly and distribution. To further support this claim we have repeated the analysis using a high resolution technique (STORM) (See also referee#2 - point 10b). Under STORM the distribution of clusters positive for Mic19/CHCHD3 in WT vs Miro DKO mitochondria can be easily observed and compared between conditions. The new images are now part of Figure 6 (Fig. 6A and B). We have applied a density-based spatial clustering (DBSCAN) and analysed distribution by calculating the Nearest Neighbour Distances (NND) which is now in panel Figure 6C. The results indicate that the clusters positive for Mic19/CHCHD3 distribute in a more uniform fashion in WT mitochondria while in Miro DKO mitochondria these clusters distribute more heterogeneously.

Further points

4. The authors should use the uniform nomenclature for the MICOS complex and its components (Pfanter et al. 2014; JCB 204:1083).

This is now rectified and uniform nomenclature according to aforementioned literature is followed throughout the revised manuscript.

5. Line 64: The authors should add the citation of Stroud et al. 2011 (*Journal of Molecular Biology*), who identified an interaction of Gem1 with ERMES in parallel to Kornmann et al.

Stroud et al. *Journal of Molecular Biology* 2011, is now cited along with Kornmann et al. *PNAS*, 2011.

6. Is GFP-Su9 (mentioned in line 108) identical with Su9-EGFP (mentioned in line 419)? Please use a uniform nomenclature for the constructs.

We apologize for the mistake and have now corrected the nomenclature of the plasmids used throughout the main text.

Also, the names and description of most plasmids are lacking. A simple reference to Addgene is not sufficient to unambiguously identify the plasmids. Please cite the original papers.

We have now checked carefully and mentioned the reference along with addgene number in the parentheses.

7. I guess, GFP-Su9 shown in Fig. 1d is soluble GFP in the matrix. Why does it show punctate structures in dSTORM images?

^{GFP}Su9 is indeed a soluble marker localized to the mitochondrial matrix. dSTORM relies on blinking from single molecules where reconstruction of images is performed by localization of each blinking spot using Gaussian fitting followed by a Gaussian blurring of the same size as that of localization precision observed during reconstruction (in our case ~15 nm for Alexa 555 and ~20-25 nm for Alexa 647 dyes). The molecular density image of the protein under investigation is then represented in the figure. Due to this localization, blurring and heterogeneous density of molecules at certain positions on the mitochondria of a soluble marker may not always appear as the homogenous staining that would be observed in a confocal image where individual molecules cannot be resolved. The punctate structures of ^{GFP}Su9 to which the reviewer has made reference are produced due to nature of the dSTORM imaging and reconstruction method used. To demonstrate that soluble ^{GFP}Su9 marker is homogeneously distributed, we carried out statistical analysis using a reported algorithm (Malkusch & Heilemann, 2016) where homogeneous and clustered appearance of a protein is identified. This analysis, known as Density-Based Spatial Clustering of Applications with Noise (DBSCAN) can identify a homogenous distribution or clustering of multiple proteins by identifying number of nearest neighbours of a protein within a specified diameter (in our case its ~20-25 nm, similar to localization precision). ^{GFP}Su9 forms a uniform domain covering the entire mitochondria (Reviewer's figures 6). In contrast, using the same analysis, Miro appeared to form nanoscale sized clusters.

Reviewer's Figure 6. Cluster analysis of ^{GFP}Su9 and ^{Myc}Miro1. After reconstruction of dSTORM images, Density-Based Spatial Clustering of Applications with Noise (DBSCAN) was performed. Briefly, the localization was sorted with a hypothetical distance of 25 nm (similar to localization precision) and any molecules present within the 25 nm radius are grouped as clusters. Note, in ^{Myc}Miro1 distinct domains of Miro are observed confirming their clustered arrangement along OMM while ^{GFP}Su9 showed homogeneous distribution. Scale bar: 1 μm.

8. The efficiency of CoIP is not clear, as it is not indicated how much input was loaded on the gels.

We have loaded approximately 2% of the lysates used for the immunoprecipitations as inputs in all immunoprecipitation experiments. We now state this in the methods section and in the figure legends.

The entire GFP blots should be shown (Fig. 2g and 4a). How many times were these experiments repeated?

Please find below the entire GFP blot for Fig. 4A (formerly Fig. 2G). For Fig. 3A (formerly Fig. 4A) we used all the pull-down samples to load one gel per experiment. This single gel was cut between the expected sizes of the GFP- and the myc-tagged Miro constructs (which differ in around 30-35 KDa). For this reason we did not produce an entire GFP western blot for this experiment. However, in the gel provided below for Fig. 4A, both ^{GFP}Miro1 and ^{GFP}Miro2 expression and pull-down can be observed and compared, which would be equivalent to the same constructs used for experiment shown in Fig. 3A. As for all immunoprecipitation experiments 2% of the immunoprecipitation volume was loaded as inputs. All immunoprecipitation experiments were performed at least 3 times.

Reviewer's Figure 7. HeLa cells transfected with GFP, ^{GFP}Miro1 and ^{GFP}Miro2 were immunoprecipitated and western blot was performed using anti-GFP antibody.

9. Fig. 5a: Is "Miro" Miro1? Please explain the arrows in the legend.

The antibody that is used in this panel was raised against Miro1 but also recognizes Miro2 (please see: (Lopez-Domenech, Covill-Cooke et al., 2018)). We have made this clear in the text when this antibody is used for the first time in Fig. 5F when detecting Miro complexes in BN-PAGE gels from WT and DKO cells. Former Fig. 5A is now Figure 1F and we have used specific Miro1 and Miro2 antibodies instead of the antibody that recognizes both proteins to avoid ambiguity.

10. Fig. 6e and f show quantifications of TEM data. It has to be stated somewhere how many cells were analyzed in how many independent experiments and whether the observed differences are statistically significant.

The experiment in Fig. 6E and F (now Fig. 1E and F in revised manuscript) were quantified from two independent experiments involving >20 individual cells from each genotype (WT and Miro DKO). This information is now mentioned in the Fig. 1 figure legends. The observed differences are significantly different as revealed by two independent statistical tests (Both One-way ANOVA ($p = 0.0198$, Tukey's post-hoc analysis) and Non-parametric two-independent samples ($p = 0.037$, Mann-Whitney test).

11. The results shown in Figs. S12 and S13 should be described in the Results section.

We have decided not to include these data in the current version of the manuscript as it was not necessary for the message we intend to deliver.

References

- Abou-Saleh H, Pathan AR, Daalis A, Hubrack S, Abou-Jassoum H, Al-Naeimi H, Rusch NJ, Machaca K (2013) Inositol 1,4,5-trisphosphate (IP3) receptor up-regulation in hypertension is associated with sensitization of Ca²⁺ release and vascular smooth muscle contractility. *The Journal of biological chemistry* 288: 32941-51
- Baumgart F, Arnold AM, Leskovaar K, Staszek K, Folser M, Weghuber J, Stockinger H, Schutz GJ (2016) Varying label density allows artifact-free analysis of membrane-protein nanoclusters. *Nat Methods* 13: 661-4
- Connolly CN, Futter CE, Gibson A, Hopkins CR, Cutler DF (1994) Transport into and out of the Golgi complex studied by transfecting cells with cDNAs encoding horseradish peroxidase. *J Cell Biol* 127: 641-52
- de Brito OM, Scorrano L (2008) Mitofusin 2 tethers endoplasmic reticulum to mitochondria. *Nature* 456: 605-10
- Fiolka R, Shao L, Rego EH, Davidson MW, Gustafsson MG (2012) Time-lapse two-color 3D imaging of live cells with doubled resolution using structured illumination. *Proceedings of the National Academy of Sciences of the United States of America* 109: 5311-5
- Harmon M, Larkman P, Hardingham G, Jackson M, Skehel P (2017) A Bi-fluorescence complementation system to detect associations between the Endoplasmic reticulum and mitochondria. *Sci Rep* 7: 17467
- Horner SM, Liu HM, Park HS, Briley J, Gale M, Jr. (2011) Mitochondrial-associated endoplasmic reticulum membranes (MAM) form innate immune synapses and are targeted by hepatitis C virus. *Proceedings of the National Academy of Sciences of the United States of America* 108: 14590-5
- Huynen MA, Muhlmeister M, Gotthardt K, Guerrero-Castillo S, Brandt U (2016) Evolution and structural organization of the mitochondrial contact site (MICOS) complex and the mitochondrial intermembrane space bridging (MIB) complex. *Biochimica et biophysica acta* 1863: 91-101
- Jans DC, Wurm CA, Riedel D, Wenzel D, Stagge F, Deckers M, Rehling P, Jakobs S (2013) STED super-resolution microscopy reveals an array of MINOS clusters along human mitochondria. *Proceedings of the National Academy of Sciences of the United States of America* 110: 8936-41
- Kornmann B, Osman C, Walter P (2011) The conserved GTPase Gem1 regulates endoplasmic reticulum-mitochondria connections. *Proceedings of the National Academy of Sciences of the United States of America* 108: 14151-6
- Kozjak-Pavlovic V, Ross K, Benlasfer N, Kimmig S, Karlas A, Rudel T (2007) Conserved roles of Sam50 and metaxins in VDAC biogenesis. *EMBO reports* 8: 576-82
- Lopez-Domenech G, Covill-Cooke C, Ivankovic D, Halff EF, Sheehan DF, Norkett R, Birsa N, Kittler JT (2018) Miro proteins coordinate microtubule- and actin-dependent mitochondrial transport and distribution. *The EMBO journal* 37: 321-336
- Lowe AR, Tang JH, Yassif J, Graf M, Huang WY, Groves JT, Weis K, Liphardt JT (2015) Importin-beta modulates the permeability of the nuclear pore complex in a Ran-dependent manner. *Elife* 4
- Malkusch S, Heilemann M (2016) Extracting quantitative information from single-molecule super-resolution imaging data with LAMA - LocAlization Microscopy Analyzer. *Sci Rep* 6: 34486
- Misko A, Jiang S, Wegorzewska I, Milbrandt J, Baloh RH (2010) Mitofusin 2 is necessary for transport of axonal mitochondria and interacts with the Miro/Milton complex. *The Journal of neuroscience : the official journal of the Society for Neuroscience* 30: 4232-40
- Nguyen TT, Lewandowska A, Choi JY, Markgraf DF, Junker M, Bilgin M, Ejsing CS, Voelker DR, Rapoport TA, Shaw JM (2012) Gem1 and ERMES do not directly affect phosphatidylserine transport from ER to mitochondria or mitochondrial inheritance. *Traffic* 13: 880-90

Niescier RF, Hong K, Park D, Min KT (2018) MCU Interacts with Miro1 to Modulate Mitochondrial Functions in Neurons. *The Journal of neuroscience : the official journal of the Society for Neuroscience* 38: 4666-4677

Owen DM, Rentero C, Rossy J, Magenau A, Williamson D, Rodriguez M, Gaus K (2010) PALM imaging and cluster analysis of protein heterogeneity at the cell surface. *J Biophotonics* 3: 446-54

Pearce MM, Wang Y, Kelley GG, Wojcikiewicz RJ (2007) SPFH2 mediates the endoplasmic reticulum-associated degradation of inositol 1,4,5-trisphosphate receptors and other substrates in mammalian cells. *The Journal of biological chemistry* 282: 20104-15

Pfanner N, van der Laan M, Amati P, Capaldi RA, Caudy AA, Chacinska A, Darshi M, Deckers M, Hoppins S, Icho T, Jakobs S, Ji J, Kozjak-Pavlovic V, Meisinger C, Odgren PR, Park SK, Rehling P, Reichert AS, Sheikh MS, Taylor SS et al. (2014) Uniform nomenclature for the mitochondrial contact site and cristae organizing system. *J Cell Biol* 204: 1083-6

Safiulina D, Kuum M, Choubey V, Gogichaishvili N, Liiv J, Hickey MA, Cagalinec M, Mandel M, Zeb A, Liiv M, Kaasik A (2019) Miro proteins prime mitochondria for Parkin translocation and mitophagy. *The EMBO journal* 38

Sengupta P, Jovanovic-Talisman T, Skoko D, Renz M, Veatch SL, Lippincott-Schwartz J (2011) Probing protein heterogeneity in the plasma membrane using PALM and pair correlation analysis. *Nat Methods* 8: 969-75

Soderberg O, Gullberg M, Jarvius M, Ridderstrale K, Leuchowius KJ, Jarvius J, Wester K, Hydbring P, Bahram F, Larsson LG, Landegren U (2006) Direct observation of individual endogenous protein complexes in situ by proximity ligation. *Nat Methods* 3: 995-1000

Takei K, Mignery GA, Mugnaini E, Sudhof TC, De Camilli P (1994) Inositol 1,4,5-trisphosphate receptor causes formation of ER cisternal stacks in transfected fibroblasts and in cerebellar Purkinje cells. *Neuron* 12: 327-42

Xu H, Guan N, Ren YL, Wei QJ, Tao YH, Yang GS, Liu XY, Bu DF, Zhang Y, Zhu SN (2018) IP3R-Grp75-VDAC1-MCU calcium regulation axis antagonists protect podocytes from apoptosis and decrease proteinuria in an Adriamycin nephropathy rat model. *BMC Nephrol* 19: 140

REVIEWERS' COMMENTS:

Reviewer #1 (Remarks to the Author):

The authors have adequately addressed my previous concerns. The manuscript is now recommended for publication.

Reviewer #3 (Remarks to the Author):

Modi et al have well commented to all of my concerns and have well justified their case especially with some new and nicely convincing experiments and arguments. I have no further concerns.

A few minor remarks.

- I have in my previous comments asked whether the TIRF-based dSTORM recordings provide good enough penetration for imaging mitochondria. Unfortunately, I did not understand the answer of the authors. How did the authors achieve a penetration depth of 1.5 μ m with TIRF? Did they use HiLo illumination – otherwise it should not be possible?

- Why do the authors not show the data of Reviewer's Figure 4 as supplementary data in the manuscript? This would really strengthen the story.

Reviewer #4 (Remarks to the Author):

The authors have significantly extended and very much improved their study. They have responded to most of my previous concerns in an adequate manner. However, I still don't find the BN-PAGEs convincing (Figs. 3F and 5F; see my previous point 2.). I recommend to remove these data from the manuscript, as they are not absolutely necessary. Moreover, the authors could mention in the discussion that they cannot exclude that altered lipid composition due to loss of ER contact sites might contribute to abnormal mitochondrial ultrastructure in the absence of Miro (see my previous point 3.).

REVIEWERS' COMMENTS:

Reviewer #1 (Remarks to the Author):

The authors have adequately addressed my previous concerns. The manuscript is now recommended for publication.

Reviewer #3 (Remarks to the Author):

Modi et al have well commented to all of my concerns and have well justified their case especially with some new and nicely convincing experiments and arguments. I have no further concerns.

A few minor remarks.

- I have in my previous comments asked whether the TIRF-based dSTORM recordings provide good enough penetration for imaging mitochondria. Unfortunately, I did not understand the answer of the authors. How did the authors achieve a penetration depth of 1.5 μ m with TIRF? Did they use HiLo illumination – otherwise it should not be possible?
- Why do the authors not show the data of Reviewer's Figure 4 as supplementary data in the manuscript? This would really strengthen the story.

We are sorry for the lack of clarity in our earlier response. As the referee points out the 1.5 μ m penetration depth was obtained from the 3D STORM microscope where Hi-Lo illumination was used. As suggested by this referee, we have now included Reviewer's Figure 4 as a Supplementary Figure. Because panel B is already a part of Figure 4 (Figure 4E) and panels C and D are equivalent to Figure 4D, we have included panel A from Reviewer's Figure 4 as part of Supplementary Figure 6 and have referenced it in the main text.

Reviewer #4 (Remarks to the Author):

The authors have significantly extended and very much improved their study. They have responded to most of my previous concerns in an adequate manner. However, I still don't find the BN-PAGEs convincing (Figs. 3F and 5F; see my previous point 2.). I recommend to remove these data from the manuscript, as they are not absolutely necessary. Moreover, the authors could mention in the discussion that they cannot exclude that altered lipid composition due to loss of ER contact sites might contribute to abnormal mitochondrial ultrastructure in the absence of Miro (see my previous point 3.).

We have followed this referee's suggestion and we have remove panels 3F and 5F from the manuscript. In addition, we have included a section in the discussion acknowledging the possible implication of altered lipid composition in the mitochondrial membranes due to loss of ER contact sites in the abnormal mitochondrial ultrastructure in Miro DKO cells.